# Competition and growth among *Aedes aegypti* larvae: Effects of distributing food inputs over time

**Kurt Steinwascher**⬢ *

Florida Medical Entomology Laboratory, Vero Beach, FL, United States of America

* kurtfsteinwascher@gmail.com

**Data Availability Statement:** AI relevant data are within the manuscript and its Supporting Information files.

**Funding:** KS-research funding and support by NIHNIAID post doctoral fellowship AI-06088 from

## Abstract

Male and female mosquito larvae compete for different subsets of the yeast food resource in laboratory microcosms. Males compete more intensely with males, and females with females. The amount and timing of food inputs alters both growth and competition, but the effects are different between sexes. Increased density increases competition among males. Among females, density operates primarily by changing the food/larva or total food; this affects competition in some interactions and growth in others. Food added earlier in the life span contributes more to mass than the same quantity added later. After a period of starvation larvae appear to use some of the subsequent food input to rebuild physiological reserves in addition to building mass. The timing of pupation is affected by the independent factors and competition, but not in the same way for the two sexes, and not in the same way as mass at pupation for the two sexes. There is an effect of density on the timing of pupation for females independent of competition or changes in food/larva or total food. Male and female larvae have different larval life history strategies. Males grow quickly to a minimum size, then pupate, depending on the amount of food available. Males that do not grow quickly enough may delay pupation further to grow larger, resulting in a bimodal distribution of sizes and ages. Males appear to have a maximum size determined by the early food level. Females grow faster than males and grow larger than males on the same food inputs. Females affect the growth and competition among males by manipulating the number of particles in the microcosm through changes in feeding behavior. Mosquito larvae appear to have evolved to survive periods of starvation and take advantage of intermittent inputs of food into containers.

## Introduction

Competition may organize communities in nature, but this is difficult to demonstrate unambiguously. Competition for food appears to affect the outcomes of laboratory microcosms [1–21]. For *Aedes aegypti* larvae in these microcosms, the food level and the density of larva both affect intraspecific competition, which shows up as the interaction between food and density. Furthermore, the total food affects the nature of competition among the larvae, also apparent

1979-1982. National Institutes of Health, National Institute of Allergy and Infectious Disease. https://www.niaid.nih.gov KS-computer time provided by Duke University Zoology Department. https://www.duke.edu KS-posthumous support from Richard Merritt. The funders had no role in study design, data collection and analysis, decision to publish, or preparation of the manuscript.

**Competing interests:** The authors have declared that no competing interests exist.

in the interaction of food and density. Density, total food, and food per larva are interdependent, but all three factors affect competition among mosquito larvae (see [1] and literature review therein). Male and female larvae compete differently from one another for yeast cells; female larvae outcompete males through larger size and by retaining cells within their gut at low total food levels. As competition intensifies, the pupal masses of both males and females decrease, so the effect of competition is a reduced apparent food level. The age at pupation is also affected by food and density. Male and female larvae delay pupation to grow larger when food is limiting, and also when food is abundant, but the cues that result in these delays differ between the sexes [1].

This report covers three experiments that investigate competition and growth of mosquito larvae in response to the distribution of food inputs during the larval life span. The first experiment covers competition among larvae, but necessarily also covers the growth of larvae. The second experiment looks at the difference between larvae grown in isolation and larvae raised with one or two competitors. Differential survival of the single larva obstructed the analysis of this data, but the survivorship results informed the design of the third experiment along with the first experiment. The third experiment extends the results of the first experiment by examining the growth of single larvae and the effect of starvation and late food additions on pupation.

The first experiment tests the hypothesis that the distribution of food inputs over time will affect intraspecific competition for food among mosquito larvae as has been shown for tadpoles [9, 22]. In nature, rainfall washes additional food into the tires and other containers that the larvae inhabit, so this is a reasonable test. Other investigators have found that food inputs affect the outcome of mosquito larval growth in microcosms [5, 7, 23–28]. Here, food and density treatments were selected to cover the span of density and food determined in a prior experiment [1] to produce significant differences in the nature of non-lethal competition among *A. aegypti* larvae. The food and density treatments should produce an interaction that will feature microcosms with the most competition, least competition and two with intermediate levels of competition. The other two factors vary the amount of food being added and the timespan over which the food is added to each microcosm. If either or both of these factors affect competition among the larvae, there should be significant 3-way or 4-way interactions among food, density, and the other two factors. The total food levels and relative food levels are high enough that interference mechanisms of competition are unlikely [1, 8, 29–33]. Individual larvae may compete exploitatively by adjusting consumption rate or the efficiency of processing food particles. Previous results suggest that males maximize consumption rate while females vary efficiency to maximize growth [1].

In addition to the hypotheses about competition, the first experiment also tests hypotheses about the growth of larvae independent of density, and about potential competition for food independent of the total food level.

The second experiment tests the hypothesis that isolated larvae grow and pupate similarly to larvae at very low densities.

The third experiment extends the investigation into the growth of mosquito larvae after a period of starvation similar to the longer delay in the first experiment. It looks at the value of late food additions and the effect of the timing and amount of those additions on the growth of the single larva. Data from the first experiment indicate that competition and differential growth results in qualitatively and quantitatively different outcomes in the treatments with longest intervals between food inputs. The third experiment tests hypotheses about the equivalence of food inputs after a period of starvation similar to those in the first experiment. There is only a single larva in each test tube, so this experiment investigates growth rather than competition.

## Methods

Greater detail about the methods and experimental design is presented in S1 Text and in dx.doi.org/10.17504/protocols.io.bddhi236.

### First experiment

Eggs were obtained from fifth generation female *A. aegypti* in a colony derived from collections in tires near the Dade County Public Works Department (Florida). Eggs were hatched in distilled water and larvae were counted into test tubes containing 20 ml distilled water. Two density treatments were crossed with two food levels treatments to produce 4 competitive environments. Two further factors, aliquot and timespan, were added to investigate the distribution of food input over time. For the aliquot factor, the total food was divided into 2 or 4 aliquots. The first aliquot was always presented on the day of hatching (day 0). The subsequent aliquots were distributed across a timespan of either 3 or 6 days. The amount and distribution of aliquots across the 16 treatments of this 4-factor experiment are listed in S1 Table. Ten biological replicates of each treatment combination were initiated. The test tubes were examined for pupae from day 4 until day 35 when the last larva died. Pupae were removed, identified to sex, and weighed to the nearest 0.01 mg. [The detailed, step-by-step methodology is presented in dx.doi.org/10.17504/protocols.io.bddhi236].

At the conclusion of the experiment, 7 dependent variables were calculated for each replicate: Survival, Prime male mass and age at pupation, Average male mass, Prime female mass and age at pupation, and Average female mass. The Prime male was the male with the greatest growth rate; this is the first male to pupate, or the largest of the first males to pupate. The Prime female was the largest female to pupate. The distinction between the two Prime individuals reflects differences in the growth and pupation of the two sexes [1, 34–36]. Test tubes with survivors of only one sex were excluded from analysis. The data were analyzed as a multivariate, four-way analysis of variance (MANOVA) [37, 38].

### Second experiment

Eggs were obtained and hatched as previously described, to produce 120 1st instar larvae. 60 flat-bottomed shell vials were filled with 20 ml distilled water containing a concentration of baker's yeast to produce the food level. 1, 2 or 3 larvae were added to each vial according to randomized treatment. 5 replicates of each treatment were initiated. S2 Table lists the treatments and replicates. The test tubes were examined for pupae from day 4 until the last larva died. Pupae were removed, identified to sex, and weighed to the nearest 0.01 mg. [The detailed, step-by-step methodology is presented in dx.doi.org/10.17504/protocols.io.bddhi236].

### Third experiment

Eggs were obtained and hatched as previously described, to produce 150 1st instar larvae. 150 test tubes were assigned to 6 treatments with 25 physical replicates each. S3 Table lists the treatments with the incremental food and delay. [The detailed, step-by-step methodology is presented in dx.doi.org/10.17504/protocols.io.bddhi236].

On day 6, after examining for dead larvae and pupae, treatments 1–3 received 1 mg, 2 mg, or 3 mg additional yeast respectively. On day 7, test tubes were examined for dead larvae and pupae. On day 8, after examining for dead larvae and pupae, treatments 4–6 received 1 mg, 2 mg, or 3 mg additional yeast respectively. On subsequent days, test tubes were examined for dead larvae and pupae. Each pupa was blotted on paper toweling to remove excess water and weighed to the nearest 0.01 mg. The weight (mg), sex (M/F), and age at pupation (days) were

recorded for each tube with a pupa. Larvae that died before pupating are also recorded by day at which death is observed.

The data are analyzed as a multivariate, three-way analysis of variance (MANOVA) [37, 38] with 3 factors: amount of incremental food (1 mg, 2 mg, or 3 mg); delay (6 or 8 days between the start of the experiment and the input of the incremental food); and sex (male or female). The two dependent variables are: mass at pupation (mg) and days to pupation after the second input of food. More detail on all three experimental designs is presented in S1 Text.

This research was conducted according to the standard guidelines at the time (1979–1982), sanctioned by the NIH, and under the supervision of the appropriate personnel at the Florida Medical Entomology Laboratory (IFAS and the University of Florida at Gainesville). Drs. J. Howard Frank and L. P. Lounibos specifically approved this study. The protocols are detailed at: dx.doi.org/10.17504/protocols.io.bddhi236

## Results and conclusions

These three experiments provide insight into how the environment affects 4 aspects of mosquito larval life: competition among mosquito larvae, the survival and growth of the larvae, and the triggers that initiate pupation. Competition, growth and the pupation triggers differ across the sexes; this suggests that survival may also differ across the sexes, but because the larvae that died were not identified by sex, this is not known. Competition also affects survival; survival decreases in the treatments with the most competition.

Detailed summaries of the three experiments and the consolidation of the results into a coherent set of conclusions are presented in S2–S5 Text (including S4–S71 Tables and S1–S38 Figs). There are more than 600 specific observations presented in the supporting information; only those that relate to the sixteen novel findings reported in the abstract are described in the main text below.

In the first experiment, the dependent variables are analyzed by multivariate analysis of variance (MANOVA); the individual analyses of variance for each dependent variable are also produced and analyzed. There are seven dependent variables for each replicate microcosm: arcsine-transformed percent Survival; the Prime male mass and age at pupation; the Average male mass at pupation; the Prime female mass and age at pupation; and the Average female mass at pupation. In addition, there are relationships between the dependent variables that are biologically important: the estimated growth rates of the Prime males and females in each treatment; and the four size distributions: among males, among females, comparing the Prime male versus the Prime female, and comparing the Average male versus the Average female. All of these are considered for each of the significant interactions between factors in the MANOVA and ANOVAs.

The four factors are food level (F), density (D), aliquot (A), and timespan (T). Each factor has two levels in the experiment (see S1 Table). The interactions between factors reveal the biological interactions among the dependent variables and are named after the contributing factors, so FxDxT refers to the interaction between food, density and timespan. This is the most important interaction in the MANOVA and one of the most important interactions in the ANOVAs. It describes the way that timespan (T) changes competition (FxD, the interaction between food and density, creates 4 competitive environments). Two other important interactions are FxT (food and timespan) and DxT (density and timespan). Both of these represent residual interactions; they describe the effect of the two factors after the competitive effects in FxDxT have been removed. FxT describes the growth of the larvae because there is no effect of density on the outcomes. DxT describes the growth of the female larvae, but competition among the male larvae; this is because density affects males and females differently. More detailed explanations are presented in S2 Text and the other supporting information.

In the third experiment, the dependent variables are also analyzed by MANOVA and ANO-VAs; the dependent variables are mass and age at pupation and the factors are the amount of the second food input, the delay between the first and second food input, and the sex of the pupa. The larvae are isolated in their test tubes so there is no competition in this experiment. The shorter delay corresponds to the longer timespan in the first experiment. More detailed explanations are presented in S2 Text and the other supporting information.

The sixteen novel findings are presented in 5 groups: competition, density, growth, timing of pupation, and life history strategies.

## Competition

**Male and female mosquito larvae compete for different subsets of the yeast food resource in laboratory microcosms.** The first experiment shows that males compete differently from females and are affected differently by the factors, aliquot and timespan. Females actively filter particles at high food levels and switch to retention at lower food levels, changing the nature of competition among females. Males appear to filter particles without switching to retention at low food levels, so they are disproportionately affected by the particle level when females switch to retention. Males appear to compete for a subset of the particles, probably because they are smaller than females in later instars.

In the FxDxT interaction (competition), in the treatments where competition is most intense among female larvae, the Prime male does better relative to the Prime female, but the non-Prime males do worse. This suggests that males and females compete for different subsets of the food resource.

In the FxA interaction (growth), all males grow largest on 4 aliquots and the effect is more pronounced at the low food level. The extra food in the third aliquot appears to contribute to the growth of Prime and non-Prime males, especially at the low food level. Prime males grow larger relative to the Average males in the 4 aliquot treatment than in the 2 aliquot treatment, and aliquot has a larger effect at the low food level than at the high food level. It appears that the third aliquot (day 2 to day 4) accelerates the growth of the Prime male. Although both Prime and Average male masses are smaller at the low food level, the Prime male in the 4 aliquot treatment is relatively large compared to both the Average male mass and the Prime female mass. This suggests that males feed on a different subset of the food resource, and that the third aliquot enables the Prime male to grow at the expense of the non-Prime males. The Prime male in this treatment is disproportionately large relative to the Average male and to the Prime female. The difference between the Prime male and Average male is due to exponential growth processes, but the difference between the Prime male and the Prime female suggests that males are using a different subset of the food particles than females.

FxT (growth) is the residual after competition has been removed; male mass appears to be directly related to the total food in the treatments with the exception of lowest total food. In the 3 treatments with the higher total food, each increment of food adds 0.2 mg to the mass of the Prime male and the Average male; this is only 40% of the increase that females realize, suggesting that males are using a subset of the food that females are using.

In the AxT interaction (growth) males grow better with 2 aliquots and the 3 day timespan than in the other treatments. The timing of the food input on day 3 accelerates the growth of males probably because the females switch from retention to actively filtering. The growth of females is similarly accelerated, but females in the 4 aliquot, 3 day timespan treatment are larger than those in the 2 aliquot, 3 day timespan. This also suggests that males are using a different subset of the food particles than the females; if females and males were using the same subsets of food particles, then they should both respond to the aliquot treatments the same way.

In the third experiment, some males pupate on day 7 or day 8 before the addition of the second food input. These males pupated on 1 mg dry weight of yeast and attained a wet mass of 1.06 mg (SD = 0.22). These males are excluded from the MANOVA analysis, but provide an additional reference point to compare the wet mass of males and females against the total dry weight of yeast in the rest of the experiment. Both males and females increase in size as the total dry weight of yeast increases. Females are larger than males at all food levels and the difference between the female wet mass and the male wet mass increases as the total dry weight of yeast increases. The slope of the 6 female wet masses plotted against the dry weight of yeast is 0.76. The slope of the 7 male wet masses plotted against the dry weight of yeast is half that, 0.38. Females appear to grow larger than males on equal dry weights of yeast. This is similar to the difference between males and females in the residual interaction FxT above.

Male larvae pupate at masses that suggest males have access to roughly half the food that female larvae feed on (FxA, FxT, third experiment). In the FxDxT interaction that describes competition, the size distribution of males and females indicates that they are competing differently in the same test tubes, suggesting that they experience different food levels. In the AxT interaction, independent of food or density, males pupate at larger sizes in the 3 day timespan treatments with 2 aliquots (day 0, day 3) than with 4 aliquots (day 0, day 1, day 2, day 3), while females pupate at larger sizes with 4 aliquots than with two. The competition interaction, supported by the growth interactions and the third experiment (growth) indicates that males and females compete for different subsets of the food particles in the experiment. The obvious distinction is the food particle size since males are usually much smaller than females.

**Males compete more intensely with males, and females with females.** Overall, food and density are the most important determinants of the outcome of competition, and both total food and food/larva affect the outcome. The size distributions of the male and female larvae are altered by competition. Non-Prime male larvae experience a small release from competition after the Prime male pupates and also grow larger on the food added on day 6 (FxDxT, FxD, DxAxT, DxT). Females are not affected by the pupation of the Prime male, but do grow larger on the food added on day 6, however this food does not appear to alter the size distribution of females as it does with males. Non-Prime females experience a release from competition with the Prime female only at the highest food level with the largest food input on day 3 (FxDxAxT).

Competition among males, among females and between males and females can be explained by active filtering and retention among females, active filtering among males, exponential growth, and delaying pupation in response to low levels of food and high levels of food, along with different life history strategies (triggers for pupation). There is no evidence for interference competition among these mosquito larvae.

Competition among females. The first experiment reveals that competition among females affects both mass and age at pupation, and is affected by both timespan and to a lesser extent aliquot, and that non-Prime females experience a release from competition with the Prime females at the highest food level with a large input of food on the third day.

In the most significant interaction describing competition, FxDxT, Prime females at high levels of food/larva grow according to the amount of total food in the test tube, while Average females grow according to the food/larva. Prime females dominate competition and the food supply. Because the Average female mass includes the Prime female mass, as the Prime female does better compared to the Average, the non-Prime females actually fare worse relative to the Prime female. The most intense competition between Prime and non-Prime females appears to be in the high food, high density, 3 day timespan treatment where the difference between the Prime and Average female mass is greatest, and the Prime female has the largest numerical advantage over the non-Prime females. At lower levels of food/larva, the Prime female changes

feeding behavior and retains food for longer times. The difference between Prime female and Average female is smaller because the nature of competition changes from active filtering, where the larger Prime female can process more food faster than the smaller females, to retention governed by the absorption rate and/or the surface area of the gut, where the advantage of the Prime female over the smaller females is not as great.

Aliquot interacts with food and density similarly to timespan (FxDxA). Within each food level, the females in the 4 aliquot treatment are larger and the difference between the Prime and Average females is smaller compared to the 2 aliquot treatment. There is intense competition between Prime and non-Prime females in the high food, high density, 2 aliquot treatment where the difference between the Prime and Average female mass is greatest, and the Prime female has the largest numerical advantage over the non-Prime females. The size of both Prime and Average females is much smaller in both most competition treatments compared to the other 6 treatments. Although the effect of the FxDxA interaction is small, it is similar to the FxDxT interaction. The addition of the third aliquot of food reduces competition for all females independently of the competition in the FxDxT interaction.

After the effects of the FxDxT and FxDxA interactions are removed, there is a residual effect of competition on the Average female mass (FxD). Because the Prime female mass is not significantly affected by this interaction, the residual effect is on the non-Prime females. These are disproportionately smaller in the most competition treatment than in the other treatments, and they are also disproportionately reduced in size relative to the Prime female in the high food, high density intermediate competition treatment. These are the same effects as seen in the two higher order interactions above, but there is a residual effect only on the non-Prime females. This suggests that the non-Prime females are more affected by competition for food than the Prime females. Females are unaffected by competition with males [1].

Competition among males. The first experiment reveals that competition among males affects both mass and age at pupation, and is affected by timespan. In the most significant interaction describing competition, FxDxT, Prime males at high levels of food/larva grow according to the amount of total food in the test tube; Prime males at intermediate and low levels of food/larva grow according to the food/larva level, but Average males do not follow either food/larva or total food. Prime males dominate competition and the food supply among males. Because the Average male mass includes the Prime male mass, as the Prime male does better compared to the Average, the non-Prime males actually fare worse relative to the Prime male. The most intense competition between Prime and non-Prime males appears to be in the high food, high density, 3 day timespan treatment where the difference between the Prime and Average male mass is greatest, and the Prime male has the largest numerical advantage over the non-Prime males. In the most competition treatment, both Prime and Average males are the smallest, but the Average male is larger than the Prime male, indicating that the non-Prime males grow larger after the Prime male pupates and on the additional food in the last food input at the end of day 6. The effect of this interaction on the non-Prime males suggests that food/larva, competition among females, release from competition after the pupation of the Prime male, and additional food on day 6 all contribute to the outcome of competition as measured in the mass at pupation. Competition among males differs from that among females and males appear to compete more intensely with other males than with females.

The FxDxA interaction has no effect on competition among males, in contrast to females. After the effects of the FxDxT interaction are removed, there is a residual effect of competition on the Average male mass (FxD). Because the Prime male mass is not significantly affected by this interaction, the residual effect is on the non-Prime males. These are disproportionately smaller in the most competition treatment than in the other treatments, and they are also disproportionately reduced in size relative to the Prime male in the high food, high density

intermediate competition treatment. In the most competition treatment, the Average male mass is again larger than the Prime male mass. In this case, there is no timespan treatment, so the difference in size reflects only the release from competition, not the addition of the final input of food on day 6. These are the same effects as seen in the higher order interaction, FxDxT, but there is a residual effect on only the non-Prime males. As with female larvae, non-Prime males are more affected by competition than Prime males. Non-Prime males experience a release from competition after the Prime male pupates and also grow larger on the final input of food in the 6 day timespan treatments. The effect of the extra food is many times larger than the effect of the release from competition (0.14 mg for food plus release in FxDxT versus 0.01 mg for just the release in FxD). There is no similar effect on the females after the Prime male pupates.

There are two other interactions that affect competition among males, DxAxT and DxT; these interactions describe the growth of females rather than competition among females. The Average male mass appears to be affected by the DxAxT interaction in the same way that it is affected by the FxDxT interaction. In this case, the size of the Average males corresponds to the estimated growth rate of the Prime male (the Prime male age and mass are not significantly affected by this interaction, but the growth rate is an indication of the competitive stress on the Prime male). The Average male grows larger than the Prime male in the treatment with the most competition (lowest growth rate). The non-Prime males grow larger than the Prime male due to the release from competition after the Prime male pupates, and on the additional food after the final input on day 6. The differences between the Prime male and the Average male are largest in the two treatments with high density and the 3 day timespan (both 2 and 4 aliquots); these are both intermediate food/larva treatments with high total food. This interaction is analogous to the FxDxT interaction with aliquot (A) replacing food level (F). Aliquot changes the competitive environment for non-Prime males independently of the interaction with food level. Other factors affect the Prime male mass, including competition (FxDxT), but the treatments that have aliquots of food added around day 3 allow the non-Prime males to grow larger. The timing of food inputs between days 2–4 appears to enhance the growth of the non-Prime males.

Analogous to the FxD interaction, there is a residual DxT interaction after the effects of FxDxT and DxAxT are removed. For all males, this appears to describe competition similarly to the FxD competition; the interaction is significant for the Prime male mass and the Average male mass. The two mass variables are in the same order as the food/larva in the test tubes. The Prime male does relatively better at the intermediate food level with more total food (high density, 3 day timespan), and the Average male mass is relatively smaller, indicating more intense competition in this treatment than in the two low density treatments. The Prime and Average males are smallest at the lowest food/larva level, and the Average male mass is larger than the Prime male mass, indicating a release from competition for the non-Prime males after the Prime male pupates and additional growth due to the final input of food on day 6. After the effects of competition in FxDxT and DxAxT are removed, there are still residual effects on the Prime and Average male masses that appear to be due to competition. The timespan treatment interacts with density analogously to the FxD interaction, but independently. The timespan treatment changes the amount of food in the test tube over time, and affects the food/larva and competition among males after the effects of FxDxT and DxAxT have been removed. This suggests that male larvae filter actively throughout their larval period and that each input of food changes the competitive environment for males.

Male and female larvae compete for different subsets of food (above) and using different feeding mechanisms (active filtering and retention for females, active filtering only for males) and appear to compete with larvae of their own sex more intensely than with larvae of the opposite sex.

**The amount and timing of food inputs alters both growth and competition, but the effects are different between sexes.** The factors, aliquot and timespan, alter the food inputs on a daily basis, and this affects both growth and competition among male and female larvae. The most obvious differences are that males and females are affected by different interactions. Females alone are affected by FxDxAxT and FxDxA (both competition), FxAxT (growth) and DxA (age at pupation). Males alone are affected by FxA (growth). Males and females are affected differently by FxDxT (competition), FxT (growth), DxAxT (competition for males, growth for females), DxT (competition for males, growth for females), and AxT (growth). In the third experiment, there are significant interactions for sex in the ANOVAs and MANOVA, and significant 3-way interactions between sex, food and delay (period of starvation), and 2-way interactions between sex and food, indicating differences in the growth of individual larvae of different sexes. The detailed examination of these interactions shows that they are due to the different feeding mechanisms, competitive effects, and responses to food inputs and starvation between the sexes.

Effect of the amount and timing of food inputs on females. In the first experiment the highest order interaction describing growth, FxAxT, is only significant for the Prime female and Average female masses at pupation. The treatments separate into three groups based on the amount of food after day 4: high food level (above 4 mg food/larva), moderate food level (4 mg food/larva), and low food level (less than 4 mg food/larva), so food level is the most important factor in this interaction. At the highest food level the largest Prime females are in the 4 aliquot treatment, but the Prime females in the 2 aliquot treatment grow faster and pupate earlier, probably because of the large input of food on day 3. Average females also grow larger on 4 aliquots than on 2 aliquots. The aliquot treatment with 4 food inputs probably allows the females to continue actively filtering for longer. A large input on day 3 allows the Prime female to switch back to active filtering and grow larger and faster than the non-Prime females, so the timing of food inputs affects the growth of females.

At the moderate food level, the Prime females grow largest and fastest in the treatment that receives 16 mg of food on day 0, followed by the 4 aliquot treatment (4 mg per day on days 0–3) and then by the 2 aliquot treatment (8 mg on day 0 and on day 3). The non-Prime females grow largest on the 4 aliquot treatment, followed by the treatment that receives 16 mg of food on day 0, followed by the 2 aliquot treatment. Prime female age at pupation is not affected by this interaction and most of the Prime females in this group of treatment pupate on day 6 before the final aliquot. The treatment that receives 16 mg of food on day 0, receives another 16 mg at the end of day 6, but this does not appear to benefit the non-Prime females. Unlike the non-Prime males that grow larger than the Prime male, the difference between the Prime female mass and the Average female mass is largest in the treatment that receives 16 mg of food on day 0. This suggests that the size distribution of the females is determined by the time of the final larval molt and that females grow as large as possible based on the amount of food, but also as limited by the size of their head capsule, feeding apparatus or some other physical feature or physiological state.

At the low food level, there is insufficient food for the Prime females to pupate before the final input on day 6. Both Prime and non-Prime females grow larger in the 4 aliquot treatment than in the 2 aliquot treatment. Non-Prime females do relatively better in the 4 aliquot treatment compared to the Prime female. This indicates that early food inputs are more valuable than later food inputs (see below) and also that multiple inputs result in better growth than a large initial input followed by a large later input. The females may switch to retention earlier in the 2 aliquot treatment and this may affect both their final molt and the efficiency of converting food to mass. Retention results in smaller females and smaller size distributions, suggesting that it is less efficient than actively filtering. Despite receiving the same total amount of food as

the 4 aliquot treatment, the females in the 2 aliquot treatment may be handicapped by the period of starvation so that the final molt does not allow them to grow as large as the females in the 4 aliquot treatment.

The FxT interaction represents the residual effect on female growth after considering the higher order interactions: FxDxAxT (competition), FxDxT (competition), and FxAxT (growth). After removing the effects of competition and growth, what remains is an almost linear relationship between the total food after day 4 and the mass at pupation for both Prime and Average females. Each incremental increase in food results in about a 0.5 mg increase in the mass of both the Prime and Average female.

The AxT interaction represents the residual effect on female growth after considering the higher order interactions: FxDxAxT (competition), FxAxT (growth), and DxAxT (growth). The four treatments in this interaction change the food distribution pattern across all food levels and densities. Females grow best in the test tubes with 4 aliquots and the 3 day timespan; this translates to the most food early in the larval period. Females do disproportionately worse in the test tubes with 2 aliquots and the 6 day timespan; the Prime females pupate almost 2 days after the final input of food, and do not grow as large as other Prime females despite the same amount of food across all treatments. The difference between the Prime female and the Average female is also largest in this treatment; this is the opposite of the pattern expected for exponential growth. There is an effect of the pattern of inputs on the growth of female larvae regardless of food or density. The disproportionately small females with the largest difference between Prime and Average masses suggests that the Prime female dominates the non-Prime females during the early larval period when food is relatively abundant, followed by a period of retention during which the Prime female maintains and slowly increases this advantage, followed by a further period of active filtering during which the Prime female extends the advantage over the non-Prime females on the second food input (day 6). The period of starvation may also differentially affect the physiological condition of the non-Prime females, further contributing to the large difference between the Prime and Average females in this treatment.

The DxAxT interaction describes competition among male larvae, but growth among female larvae. Only the Prime female mass is affected by this interaction; the non-Prime females must be reduced in size collectively as the Prime female increases in size in order for the Average female mass not to be affected by this interaction. There are different effects at the higher and lower food levels (after day 4). At the highest food level, the Prime females in the low density, 2 aliquot, 3 day timespan treatment are larger than those in the low density, 4 aliquot, 3 day timespan treatment; the main effects predict the opposite. The timing of the large second food input on day 3 accelerates the growth of the Prime female at the expense of the non-Prime females. In the low density, 4 aliquot, 3 day timespan treatment, the Prime females are smaller, but the difference between the Prime and Average females is also smaller; the 4 aliquots favor the growth of the non-Prime females relative to the Prime females.

In the 6 treatments at lower food levels, the mass of the Prime female follows the food/larva after day 4; where the food/larva is equal across different treatments, the total food determines the larger female, and in the two treatments where both food/larva and total food are equal, the treatment with 4 aliquots is larger than the treatment with 2 aliquots. In the high density, 2 aliquot, 6 day timespan treatment the masses of the Prime female and Average female are smallest and the difference between the Prime and Average female masses is greatest. This treatment has the lowest level of food/larva.

At both the highest food level and the lowest food level the Prime female benefits from the large second aliquot relative to the non-Prime females. This is related to density, but not competition. At the low density, the Prime female benefits over the non-Prime females from the large aliquot on day 3. At high density, the Prime female benefits over the non-Prime females

from the large aliquot on day 6. Food/larva rather than density, aliquot and timespan appears to explain the size of the Prime female at intermediate food levels. This suggests an effect of density (independent of competition) that interacts with the distribution of food inputs during the larval lifespan of females. The most likely explanation is that the Prime female switches from retention to active filtering with the introduction of the second aliquot in these two treatments and grows more rapidly than the non-Prime females. That the Average mass is unaffected suggests that the large input on day 3 is unevenly divided between the Prime and non-Prime females, but results in the same Average mass (for instance, across the 2 aliquot and 4 aliquot treatments at low density and the 3 day timespan). The longer period of starvation (6 days rather than 3 days) affects the distribution of sizes (and perhaps physiological states) among the female larvae so that when food is added on day 6 the Prime female is already at a greater advantage relative to the non-Prime females. The switch from retention to active filtering allows the Prime female to grow even larger relative to the non-Prime females.

In other interactions the Prime female dominates competition based on the total food at the higher food/larva levels, and follows the food/larva at lower levels, altering the size distribution of the female larvae as a result. There is no evidence for competition in this interaction, but low density results in unequal growth at high food levels (2 aliquots, 3 day timespan versus 4 aliquots, 3 day timespan); there is also unequal growth at high density and the lowest food level (2 aliquots, 6 day timespan). The timing of the input on day 3 probably coincides with the peak growth of females, while the delay to day 6 results in 3$^{rd}$ and 4$^{th}$ instar larvae that are stunted and can not grow as large as females in test tubes with more food earlier in the larval life.

The DxT interaction represents the residual effect on Prime female mass and Average female mass after the higher order interaction effects are removed. These interactions include competition (FxDxT, FxDxAxT) and growth (DxAxT). The residual effect on male larvae appears to be competition, but the residual effect on females appears to affect growth. There is an almost linear correspondence between the food/larva after day 4 and the masses at pupation for both Prime and Average females. Despite the low food/larva levels in some treatments, the ongoing food inputs over the timespan appear to allow the females to actively filter particles. The incremental increases in food across these four treatments are not regular and the corresponding increases in Prime and Average female mass are also not regular, but the slope of the line through these points is about half that of the FxT residual.

In the third experiment, individual larvae receive 1 mg of food on day 0 and then 1 mg, 2 mg, or 3 mg of food on day 6 or on day 8. Females do not pupate on less than 2 mg of food (total), which is consistent with the 2 mg food/larva necessary in the first experiment (16 mg total food for 8 larvae). Females grow larger in response to the amount of the second food input. For the shorter period of starvation each incremental increase in the food input increases mass at pupation. For the longer period of starvation (the day 8 delay) females grow larger on each additional increment of food, but not as large as in the comparable day 6 delay treatment. Each additional increment of food contributes less to the mass at pupation, and the longer period of starvation further reduces the contribution of each increment to the pupal mass. Female larvae in this experiment appear to be food limited at the 2 mg second food input and perhaps even at the 3 mg food input.

Effect of the amount and timing of food inputs on males. In the FxA interaction (growth), Prime males grow larger in the 4 aliquot treatment than in the 2 aliquot treatment and aliquot has a larger effect at the low food level than at the high food level. It appears that the third aliquot (day 2 to day 4) accelerates the growth of the Prime male. Although both Prime and Average male masses are smaller at the low food level, the Prime male in the 4 aliquot treatment is relatively large compared to both the Average male mass and the Prime female mass. This

suggests that the third aliquot enables the Prime male to grow at the expense of the non-Prime males. This is the result of exponential growth processes rather than competition. The Average male masses are in the same order as the Prime male masses, so all males appear to benefit from the 4 aliquot treatment. The Average male mass in the low food, 2 aliquot treatment is the smallest across this interaction, but it is the same as the Prime male mass, so the non-Prime males grow as large as the Prime males (probably on the food added on day 6 in some of the test tubes). There is no release from competition in this interaction, but there is additional food in some test tubes. Despite the late input of additional food, the non-Prime males don't grow as large as the non-Prime males in the low food, 4 aliquot treatment (equivalent food, delivered earlier in the larval life). One possible mechanism could be that food delivered earlier in the larval period results in larger 3rd and 4th instars and these larvae are able to grow larger than peers that did not receive the early food inputs. Additional food late in the larval life allows growth, but that growth is capped because of physiological or physical limitations (size of the feeding apparatus or other body parts).

In the FxT interaction (growth), the residual effect on female growth is an almost linear relationship between the total food after day 4 and the pupal mass of females. Prime male mass and Average male mass are also directly related to the total food in three of the four treatments (0.2 mg increments versus 0.5 mg increments for the females). The Prime male at the lowest total food is disproportionately small, but still pupates before the final input on day 6. The Average male mass in this treatment is larger than the Prime male mass, so the non-Prime males grow larger than the Prime male on the final input of food on day 6. Competition has been removed from this interaction (by FxDxT) so this represents only the effect of additional food on day 6. The reason behind the disproportionately small size for the males in the low food, 6 day timespan treatment is probably the retention of food particles by the females reducing the apparent food level for the males. The Prime male pupates before the addition of food on day 6; the non-Prime males and females grow larger on the day 6 food input, but do not grow as large as larvae that received the same amount of food earlier.

In the AxT interaction, the residual effect on females indicates that more food early in the larval life increases the mass for both Prime and Average females. There are no higher order interactions for the Prime male mass and age, and only DxAxT for the Average male mass. Both Prime and Average males grow disproportionately largest in the 2 aliquot, 3 day timespan treatment and disproportionately smallest in the 2 aliquot, 6 day timespan treatment. The two 4 aliquot treatments are intermediate in size, but the 2 aliquot, 3 day timespan treatment should be smaller than the 4 aliquot, 3 day timespan treatment (based on main effects), so a large input of food on day 3 benefits all males, perhaps because females switch from retention back to active filtering on the food input. The 4 aliquot, 6 day timespan treatment has 3/4ths the food after day 4 that the two 3 day timespan treatments have, but the males grow almost as large as the males in the 3 day timespan treatments. The multiple food inputs in this treatment cause the females to switch back to active filtering and this benefits the growth of males by increasing the number of particles in the test tubes. The worst treatment for both males and females is the 2 aliquot, 6 day timespan, where females retain food during most of the larval period and reduce the mass of males as well as that of females. After the food input on day 6, the females switch back to actively filtering; this allows the non-Prime males to grow larger on the day 6 food input and approach the mass of the Prime male in this treatment. The non-Prime males also appear to grow larger on the day 6 food input in the 4 aliquot treatment and approach the larger mass of those Prime males. The growth and mass at pupation of males in this interaction makes sense in the context of the growth and feeding behavior of the female larvae.

In the third experiment, the masses at pupation of males are affected by the period of starvation (delay treatment) and the amount of food in the second food input, but differently than

for females. At the shorter delay, equivalent to the 6 day timespan treatment in the first experiment, males grow larger as the second food input increases. At the longer delay (8 days, 2 extra days of starvation), males grow larger than at the shorter delay. Males at the longer delay and 2 mg second food input, and at both delays and the 3 mg second food input, pupate at 2.27 mg, which appears to be the maximum size of males in this experiment. Because this maximum does not seem dependent on the size of the second food input, it must be determined by environmental conditions before day 6. Males do not appear to be food limited at the 3 mg food input and may not be food limited even at the 2 mg food input.

In addition to the different interactions describing competition and growth for male and female larvae, and the differences in competition described earlier, the amounts and timing of the food inputs affect the growth of the two sexes differently. At high food levels, the growth of the Prime female is increased by the third aliquot of food, while at the lowest food level the non-Prime females appear to benefit from the 4 aliquot treatment relative to the Prime female. At intermediate levels of food, a large initial input benefits the Prime female over the non-Prime females; non-Prime females do best with the 4 aliquot treatment. The residual effect of food results in a linear increase in mass of females in the first experiment. In the third experiment, each incremental increase also results in larger females, but additional increments contribute less to the mass at pupation, and the longer period of starvation reduces the mass at pupation further.

Males do not appear to be affected by the size of the initial food input, but benefit from a large input on day 3. Prime males grow relatively larger than non-Prime males at the low food level delivered in 4 aliquots. Non-Prime males at low food levels and intense competition may grow as large or larger than Prime males in the same treatments in response to late food additions on day 6. Males in the AxT interaction grow better on the 2 aliquot, 3 day treatment relative to females. Males in the FxT interaction and the third experiment grow at a lower rate than females on the same amount of food. In the third experiment, males grow larger after the longer delay, but also reach a maximum size. Males appear to be affected by female feeding behavior (retention versus active filtering) as much as by the food level in these interactions.

## Density

**Increased density increases competition among males.** Density affects competition among males; all the significant interactions involving density describe competition among males (FxDxT, FxD, DxAxT, DxT). The factors: food (F), aliquot (A), and timespan (T), interact with density (D) to alter the competitive environment for male larvae. Higher density increases competition among males, reducing the mass of males at pupation, and increasing the age at pupation for Prime males. There are additional residual main effects of density on the masses of male pupae. Higher density reduces the pupal mass of all males after the effects of competition have been considered.

**Among females, density operates primarily by changing the food/larva or total food; this affects competition in some interactions and growth in others.** Density operates on female mass either through competition (FxD set of interactions) or through altering the food level (the non-FxD interactions describing growth). Density affects competition among females and the growth of females by altering the food/larva and/or the total food in the test tube. At the higher density, the food/larva is lower at both food levels; this increases competition among females and reduces the growth of the Prime and Average females. In the FxDxT interaction (competition) the Prime females grow as though the total food is available to them at the higher levels of food/larva, but grow only as well as the food/larva indicates at lower levels. The Average females in this interaction grow according to the food/larva at all levels of

food. In the FxDxA and FxDxAxT interactions (both competition), the size of the Prime and Average females is directly related to the food/larva. In the FxT, DxAxT and DxT interactions (all growth) the residual effect of the factors link food level to the mass at pupation. There is no residual main effect of density on female mass at pupation after the interactions with food, aliquot and timespan have been removed. This means that there is no effect of density on the mass of females after the interactions have been accounted for. The only effect of density is through the food/larva or total food.

## Growth

**Food added earlier in the life span contributes more to mass than the same quantity added later.** Effect of early food on the mass of females. There are three comparisons that support this statement: the effect of 4 aliquots versus 2 aliquots (more food earlier in the larval life); the difference between food added on day 3 versus day 6 (the effect of timespan, more food before day 4); and the incremental food additions on day 6 and day 8 in the third experiment.

The pupal mass of female larvae in the first experiment is determined by the amount of food after day 4; this quantity is affected by all four factors in various interactions. FxT and DxT reveal that the amount of food after day 4 is linearly related to the size of Prime and Average females after the higher order interactions (both competition and growth) are removed. FxAxT and DxAxT indicate different effects of food on growth at different levels of food after day 4. At the highest food level in both interactions, the addition of a large amount of food on day 3 (2 aliquot treatment) accelerates the growth of the Prime female over the non-Prime females; the non-Prime females do better in the 4 aliquot treatment. At the lowest food level in both interactions, there is not enough food for females to pupate until after the final food input on day 6. Females grow larger on the 4 aliquot treatment than the 2 aliquot treatment because there is more food by day 4, but the large input on day 6 results in a larger difference between the Prime and Average female masses as a result of exponential growth processes and active filtering of the additional food particles. This indicates that early food inputs are more valuable than later food inputs and also that multiple inputs result in better growth than a large initial input followed by a large later input. The females may switch to retention earlier in the 2 aliquot treatment and this may affect both their final molt and the efficiency of converting food to mass. Retention results in smaller females and smaller size distributions, suggesting that it is less efficient than actively filtering. Despite receiving the same total amount of food as the 4 aliquot treatment, the females in the 2 aliquot treatment may be handicapped by the period of starvation so that the final molt does not allow them to grow as large as the females in the 4 aliquot treatment.

The AxT interaction is the residual for these two factors after competition and growth have been removed. Early food inputs result in better growth and the 2 aliquot, 6 day timespan results in disproportionately smaller females across all food levels and densities.

In the third experiment, individual larvae receive 1 mg of food on day 0 and then 1 mg, 2 mg, or 3 mg of food on day 6 or on day 8. Females do not pupate on less than 2 mg of food (total), which is consistent with the 2 mg food/larva necessary in the first experiment (16 mg total food for 8 larvae). The mass at pupation is affected by both the amount and timing of the second food input in this experiment. Females grow larger in response to the amount of the second food input. Each additional increment of food contributes less to the mass at pupation, and the longer period of starvation further reduces the contribution of each increment to the pupal mass. For the shorter period of starvation each incremental increase in the food input increases mass and decreases the age at pupation. For the longer period of starvation (the day

8 delay) females grow larger on each increment of added food, but not as large as in the comparable day 6 delay treatment, and they take longer to grow that large. Females appear to have a maximum larval period that causes them to pupate, resulting in smaller pupae despite equivalent amounts of food. Female larvae in this experiment may be food limited at the 2 mg second food input and perhaps even at the 3 mg food input.

Equal increments of food have different values to female growth depending on the level of food already present and the timing of the input during the larval lifespan. Early food inputs are more valuable to growth than later food inputs. Inputs during the day 2 to day 4 period at high food levels have a larger effect on growth than earlier or later inputs. Each input appears to cause females to switch from retention to actively filtering, or to continue actively filtering. Active filtering results in larger females and larger size distributions among females than retention. Incremental food inputs after a period of starvation (the day 6 or day 8 delay) result in larger females, but each additional increment contributes less to the mass of the females and the longer starvation period also reduces the value of each increment.

Effect of early food on the mass of males. Male larvae are also affected by the aliquot and timespan treatments in the first experiment and the incremental food additions in the third experiment, but the effects of these treatments are different for males than for females. The mass and age at pupation of males are more affected by competition than by growth; all interactions involving density appear to describe competition among males. There are only three interactions that describe the growth of male larvae: FxT, FxA, and AxT.

FxT is the residual after competition has been removed; male mass appears to be directly related to the total food in the treatments with the exception of lowest total food. In the 3 treatments with the higher total food, each increment of food adds 0.2 mg to the mass of the Prime male and the Average male. The males in the low food, 6 day timespan treatment are disproportionately smaller than the males and females in other treatments and the females in this treatment. The Prime male pupates before the addition of food on day 6; the non-Prime males and females grow larger on the day 6 food input, but do not grow as large as larvae that received the same amount of food earlier. The non-Prime males in the low food, 6 day timespan treatment grow larger than the Prime males on the additional food added on day 6.

Males grow largest on 4 aliquots in the FxA interaction and the effect is more pronounced at the low food level. The extra food in the third aliquot appears to contribute to the growth of Prime and non-Prime males, especially at the low food level. The Prime male in the low food, 4 aliquot treatment is disproportionately large relative to the Average male. The difference between the Prime male and Average male is due to exponential growth processes. In the low food, 2 aliquot treatment, despite the late input of additional food the non-Prime males don't grow as large as the non-Prime males in the low food, 4 aliquot treatment (equivalent food, delivered earlier in the larval life). Some non-Prime males grow larger on the additional food after day 6 (in some treatments), but the food added on day 6 is not as beneficial as food added earlier.

In the AxT interaction males grow better with 2 aliquots and the 3 day timespan than in the other treatments. The timing of the food input on day 3 accelerates the growth of males probably because the females switch from retention to actively filtering. Males do least well in the treatment with 2 aliquots and the 6 day timespan. Females retain food during the period of starvation between day 0 and day 6 and all males are smaller. The non-Prime males in both the 6 day timespan treatments grow almost as large as the Prime males in those treatments on the food input on day 6. The males in the 2 aliquot, 6 day timespan treatment don't grow as large as the males in the 4 aliquot, 6 day timespan treatment despite equivalent food; the earlier food inputs in the 4 aliquot treatment increase the mass of all males compared to the equivalent food delivered later (the 2 aliquot treatment).

In the third experiment, males grow larger on the day 8 delay than on the day 6 delay, but take longer to grow that large. This appears to be related to differences in the life history strategies between males and females, and to the differences in the response to starvation between the sexes.

Despite differences across the two sexes, food added earlier in the life span contributes more to the pupal mass than equivalent amounts added later for both sexes.

**After a period of starvation larvae appear to use some of the subsequent food input to rebuild physiological reserves in addition to building mass.** Both mass and age at pupation may be affected by the physiological state of the larvae (both sexes); this is needed to explain the differences observed at different food inputs after the two periods of starvation (in the third experiment), and also in the FxAxT interaction. In the FxAxT interaction (growth), at the low food level, there is insufficient food for the Prime females to pupate before the final input on day 6. The Prime females in the 2 aliquot treatment take almost 2 days longer to pupate than those in the 4 aliquot treatments suggesting that increasing mass alone is not the reason behind the timing of pupation after the period of starvation.

In the third experiment, the amount of food added after a period of starvation (6 days or 8 days) affects the mass at pupation of females. The size of the female increases as the amount of food increases, but each increment of food contributes less to the pupal mass. The longer period of starvation also reduces the value of the food to the pupal mass, and also decreases the value of each subsequent increment. It appears that females use food inputs after a period of starvation to improve physiological status as well as to increase mass. The amount of food added after a period of starvation also affects the mass at pupation of males. Similar to female larvae, the size of the male increases as the amount of food increases, and each increment of food contributes less to the pupal mass. Unlike females, the longer period of starvation also results in larger males. Because not all the food in the second food input results in growth, it appears that males also use food inputs after a period of starvation to improve physiological status as well as to increase mass.

In the absence of sufficient food to complete pupation, both males and females wait for additional food inputs. Both males and females pupate at a larger size and earlier as the amount of the late food input increases, but the length of the period of starvation affects the value of the food and the contribution to pupal mass. Males grow larger and take longer to pupate after the longer period of starvation compared to the shorter (6 day) period. Females do not grow as large after the longer period of starvation and appear to be limited to a specific interval of growth after the late food input.

Both males and females appear to use some of the second food input to replenish internal physiological factors after a period of starvation; the pupal mass does not reflect the entire value of the late food input and this is more apparent after the longer period of starvation.

## Timing of pupation

**The timing of pupation is affected by the independent factors and competition, but not in the same way for the two sexes, and not in the same way as mass at pupation for the two sexes.** The age at pupation is affected differently by competition than the mass at pupation in both sexes, and the effect on age at pupation is different between the sexes. The Prime female age is inversely related to the total food in the test tube while the Prime male age at pupation is not. The Prime male age is earlier, more simultaneous, and the small deviations appear to be due to food scarcity or food abundance. There is a residual effect of competition on age at pupation for both sexes after the higher order interactions have been removed; increased competition delays pupation for both the Prime female and the Prime male without affecting either mass. The interactions affecting growth of larvae also affect the age at pupation of males and females differently.

Timing of pupation for females. In the first experiment (FxDxT, competition), the Prime female age at pupation is inversely related to the total food in the test tube at all levels of food/larva after day 4. The Prime female age at pupation is not affected by the FxDxA interaction. There is also a residual effect on the Prime female age at pupation (FxD); this increases disproportionately with increasing competition after the higher order interactions have been removed. The mass at pupation of the Prime female is determined by the higher order interactions, and by the main effects, but the age at pupation increases with increasing levels of competition in this residual interaction.

The DxA interaction is only significant for Prime female age at pupation. The age at pupation is inversely related to the food/larva after day 4, but the Prime female pupates disproportionately early in the low density, 2 aliquot treatment. In contrast to interactions including timespan, the difference across the DxA treatments appears to be the amount of food in the initial input. Prime females in the low density, 2 aliquot treatment have more food/larva on day 0 than the other treatments; this accelerates their growth and results in the early pupation relative to the expected values.

The Prime female age at pupation in the DxT interaction follows the total food after day 4 rather than the food/larva, after the effect of the FxDxT interaction is removed. This is similar to the age at pupation in the FxDxT interaction (competition), but the DxT interaction appears to describe exponential growth processes for the female mass variables. For the Prime female, age at pupation may be more affected by the food environment than by competition.

Female age at pupation is affected by the amount of food (total food, initial food on day 0). Females extend their larval period until there is sufficient food for them to pupate (16 mg for 8 larvae in the first experiment, 2 mg food/larva in the third experiment). In the third experiment, females further delay pupation in response to a period of starvation. Despite the additional time, the females don't grow as large as females that received food earlier, suggesting that some of the delay is to rebuild internal physiological resources in addition to increasing mass. Females also appear to have a maximum larval period that is affected by late food inputs after starvation (in the third experiment).

Timing of pupation for males. The Prime male age at pupation is affected by competition (FxDxT, FxD, DxT), but is not in the same order as either the total food in the test tube, the food/larva, or the Average male mass. 7 of the 8 treatments (FxDxT) pupate early (between 5.0 and 5.12 days), but the Prime males in the most competition, 6 day timespan treatment pupate later, at 5.70 days. In two cases, Prime males with greater access to food delayed pupation slightly, perhaps to increase in size or improve physiological status. For males, age and mass at pupation are affected differently by competition and the factors: food, density and timespan. Males pupate earlier than females, and at a smaller size, and extend their larval life in response to too little food (most competition), and when food is relatively abundant. There is also a residual effect on the Prime male age at pupation (FxD); this increases disproportionately with increasing competition after the higher order interactions have been removed. The mass at pupation of the Prime male is determined by the higher order interactions, and by the main effects, but the timing of pupation is increased by increased levels of competition in this residual interaction. In the DxT interaction, the Prime male delays pupation in the two treatments with the most total food, and in the treatment with the least food/larva and most competition. This is consistent with observations in other interactions, especially FxDxT. Non-Prime males are more affected by competition than Prime males and pupate later than Prime males.

The Prime male age at pupation is also significantly affected by three interactions describing the growth of males: FxA, FxT and AxT. In the FxA interaction, the Prime males pupate earlier at the high food level than at the low food level, but grow larger and pupate later in the 4 aliquot treatment rather than the 2 aliquot treatment at both food levels. Prime males delay pupation in

the 4 aliquot treatment disproportionately longer at the lower food level to grow larger. It appears that the third aliquot of food (day 2 to day 4) accelerates the growth of the Prime male, affecting the timing of pupation as well as the mass. In the FxT interaction, Prime males delay pupation slightly at the highest food level and more at the low food levels, with the longest delay occurring at the lowest food level. They grow for longer and reach a larger size when food is most available, and also delay pupation when food is less available, but they still pupate on day 5 or day 6. In the AxT interaction, the Prime males pupate earlier in the treatments with the 3 day timespan (food delivered earlier in the larval life). Prime males pupate earliest in the 4 aliquot, 3 day timespan treatment (in which females grow largest) and latest in the 4 aliquot, 6 day timespan treatment, probably due to the third aliquot food input on day 4. The Prime male pupates earlier than expected in the test tubes where the females grow largest and fastest, and later than expected in the test tubes where additional food is added late in the larval period (on day 4).

In the third experiment, the age at pupation of males is affected by the period of starvation (delay treatment) and the amount of food in the second food input, but age at pupation is affected differently than for females. At the shorter delay, equivalent to the 6 day timespan treatment in the first experiment, males grow larger and pupate earlier as the second food input increases. At the longer delay (8 days, 2 extra days of starvation), males grow larger than at the shorter delay, but take longer to pupate.

Males delay pupation to grow larger when food is most abundant. They also delay pupation to grow larger when food is scarce, especially in test tubes where females are retaining food. Prime males pupate on day 5 or day 6, but some non-Prime males delay pupation until after day 6 and grow larger on the final input of food (especially at the lowest food levels). The timing of pupation for males is affected by the food level, late inputs of food (day 2 through day 4), starvation, and the feeding behavior of females. In contrast to the females, males delay pupation when food is abundant and when food is scarce, and non-Prime males delay pupation to grow larger after the Prime males pupate.

**There is an effect of density on the timing of pupation for females independent of competition or changes in food/larva or total food.** The age at pupation of Prime females is altered by the interaction DxA. The exclusion of food level and timespan from this interaction suggests that density and aliquot have an effect separate from total food or food/larva. At low densities, both aliquot treatments pupate early, but at high densities, the 2 aliquot treatment is much later than the 4 aliquot treatment. The source of the interaction appears to be that the low density, 2 aliquot treatment pupates much earlier than projected by the main effects. This is probably due to the large initial input of food; the food/larva on day 0 for this treatment is 2 mg or 4 mg, depending on the food level, twice that of the next highest initial food levels. This only affects the timing of pupation. The Prime female grows to a size and at a rate commensurate with the main effects and other interactions, but pupates earlier than expected because of the large initial food input.

There is also a residual effect on the age at pupation of Prime females in the DxT interaction. Age is inversely related to the total food in the treatments; this is the same relationship as in the FxDxT interaction for which this is one of the residuals.

There is no residual effect of density on the masses of either the Prime or Average females, but there is a large residual effect of density on the age at pupation of Prime females. Increased density increases the age at pupation of the Prime females.

## Male and female larvae have different larval life history strategies

In this context, life history strategies refer to the collective responses that the larvae make to the food environment in these microcosms. In nature, there are other factors affecting the

outcomes, but in these experiments, only the food environment and density change across treatments. Male and female larvae consistently differ in their growth rates, in their responses to food shortages and abundance, in the various factors that trigger molts and pupation, in the efficiency of feeding, and in the character of their feeding behavior. Both sexes appear to be adapted to conditions of starvation during the larval period, although their responses to starvation are different.

**Males grow quickly to a minimum size, then pupate, depending on the amount of food available.**   Males appear to have a minimum mass determined by the food environment above which pupation can occur. Evidence for this is at the lowest food levels in the interactions that describe the growth of males: FxA, FxT and AxT. The FxA interaction is only significant for males in the ANOVAs. The mass of the Prime male corresponds to the amount of food in the treatments by day 4. Prime males pupate before day 6. Some non-Prime males in the FxA interaction do grow larger on the day 6 food input and the Average male mass is the same as the Prime male mass at the low food level, 2 aliquot treatment (the least amount of food on day 4). This suggests that there is a minimum mass at pupation for males that is determined by environmental conditions around day 4 (the day the third aliquot is added for half the timespan treatments). Despite receiving much more food on day 6, the non-Prime males at the low food level, 2 aliquot treatment do not grow as large as the non-Prime males in the 4 aliquot treatment at the same food level (delivered earlier). Males grow until they reach the minimum mass determined on day 4 (the Prime males), then grow larger if there is abundant food, delaying pupation in order to grow (the non-Prime males).

The FxT interaction shows this pattern as well. The non-Prime males in the low food, 6 day timespan treatment grow larger than the Prime males (0.04 mg larger), but not as large as the non-Prime males in the low food, 3 day timespan treatment (same amount of food, delivered earlier). The extra food benefits the non-Prime males, but not as much as the same amount of food added on day 3.

The AxT interaction also shows the pattern. The non-Prime males in the 2 aliquot, 6 day timespan treatment grow larger than the Prime males (0.02 mg larger), but not as large as the non-Prime males in the 4 aliquot, 6 day timespan treatment (with the next higher food level on day 4). In this interaction the third aliquot of food on day 4 delivers the incremental food, which appears to change the minimum size of the male pupae.

One possible mechanism could be that food delivered earlier in the larval period results in larger 3$^{rd}$ and 4$^{th}$ instars and these larvae are able to grow larger than peers that did not receive the early food inputs. Additional food late in the larval life allows growth, but that growth is capped because of physiological or physical limitations (size of the feeding apparatus or other body parts).

**Males that do not grow quickly enough may delay pupation further to grow larger, resulting in a bimodal distribution of sizes and ages.**   The age at pupation of Prime male larvae is affected by the interactions describing growth; these differ from the interactions that affect the age at pupation of Prime female larvae (DxA and DxT). For Prime males, the growth interactions FxA, FxT, and AxT, are significant for age at pupation. Prime males delay pupation when food is abundant, and also when food is scarce. This is the same response that Prime males exhibit in the competition interactions.

In addition to the pupal triggers based on mass, there appear to be different triggers based on the age of the larvae. Males pupate earlier than females, and for Prime males the distribution of ages is much tighter than for Prime females. All Prime males pupate by day 6. The age and mass triggers are not independent of each other; both Prime males and Prime females appear to extend the larval period at low food levels in order to grow larger. Prime males also appear to delay pupation at high food levels in order to grow larger. Non-Prime males

sometimes extend their larval period to grow larger than the Prime male; this suggests an alternate life history strategy for non-Prime males based on growing to a larger size (increased adult longevity) rather than early emergence.

**Males appear to have a maximum size determined by the early food level.** Males appear to have a maximum mass at which pupation occurs, also determined by the food environment early in the larval period. The difference in size between the Prime male and the Prime female increases as the size of the Prime female increases in the first experiment. This means that the Prime male reaches some limit based on the amount of food in the test tube and pupates while the Prime female continues to grow. This is even more apparent in the third experiment. Males at the longer delay and 2 mg second food input, and at both delays and the 3 mg second food input, pupate at 2.27 mg, which appears to be the maximum size of males in this experiment. Because this maximum does not seem dependent on the size of the second food input, it must be determined by environmental conditions before day 6.

**Females grow faster than males and grow larger than males on the same food inputs.** In the first experiment, females consistently pupate at a larger size than males in the same treatment. Females at higher food levels are disproportionately larger than males at those food levels (see also the maximum size of males, and males using a different subset of the food resources, above). The growth rates of males and females also show larger differences at higher food levels (lower competition), although even at the lowest food levels and most competition treatments, the female growth rate is higher than the male growth rate. In the FxT interaction (growth), the residual effect of food and timespan is a linear relationship between the size of the pupae and the amount of food (after day 4); females pupate at double the mass at which males pupate.

In the third experiment, males and females increase in size as the total dry weight of yeast increases. Females are larger than males at all food levels and the difference between the female wet mass and the male wet mass increases as the total dry weight of yeast increases. The slope of the female wet masses plotted against the dry weight of yeast is double that of the male wet masses (0.76 versus 0.38). Females grow larger than males on equal dry weights of yeast. This is similar to the difference between males and females in the residual interaction FxT.

**Females affect the growth and competition among males by manipulating the number of particles in the microcosm through changes in feeding behavior.** Competition among males and the growth of males is affected by the availability of food particles, which is largely controlled by the female larvae. When female larvae experience low levels of food particles, they switch to retention, further reducing the level of food particles. Male larvae appear to actively filter particles throughout their larval life. This difference between males and females is apparent in the interactions describing competition for males (FxDxT, FxD, DxAxT, DxT) where the males are disproportionately small in the treatments with the least food and most competition, and where the females exhibit the smaller size distribution characteristic of retention. It is also apparent in the FxT interaction describing growth for all larvae; the Prime male is disproportionately small, but the non-Prime males grow larger than the Prime male on the added food on day 6. It is less apparent in the AxT interaction (growth); the Prime male is also disproportionately small, but the non-Prime males do not quite grow as large as the Prime males in the treatment with the least food.

**Mosquito larvae appear to have evolved to survive periods of starvation and take advantage of intermittent inputs of food into containers.** Competition in the first experiment has a small effect on survival (FxDxT accounts for 5% of the variance), but the distribution of the food in time (timespan treatment, FxT and DxT interactions) accounts for 10% of the variance. Survival is better in the treatments where the higher amounts of food are added over the longer timespan. Survival may be negatively affected by higher amounts of food at the shorter

timespan. In the second experiment, too much food appears to reduce survival at very low densities (1 or 2 larvae per vial). Too little food reduces survival, but too much food too early in the larval period also reduces survival. (see also [1, 39])

The aliquot treatment in the first experiment also affects survival; larvae survive better on 2 aliquots than on 4 aliquots (12% of the variance). This is opposite of the effect of the aliquot treatment on mass and age at pupation (larvae grow better on 4 aliquots than on 2 aliquots). There are no interactions between aliquot and the other treatments for survival; the effect of aliquot is independent of the interactions described previously. The 4 aliquot treatment provides more food early in the larval period than the 2 aliquot treatment, so this result is symmetrical to the observations above that too much food too early in the larval period reduces survival, but the effect of the number of aliquots is independent of food, density, competition, and timespan.

Overall, the survival observations suggest that mosquito larvae are adapted to low levels of food and conditions of starvation, and that these experiments provide food levels that may be at the high end of those encountered in nature.

## Conclusions

The change in size due to exponential growth processes looks like a sigmoid curve when plotted against time. For mosquito larvae each instar should resemble a sigmoid curve. Without having measured the growth trajectories of each instar, the data from these experiments is consistent with female larvae growing to the top of the sigmoid curve and male larvae molting and pupating at some lower level. This takes into account the data on growth rates, and masses and ages at pupation. Females grow larger because at each molt, they attain a larger size than the males in the cohort. They grow faster because they molt/pupate at a point higher on the sigmoid curve than the males, and the line between the origin for that instar and the point at which the molt/pupation occurs will have a greater slope (growth rate) for females than males, all other conditions being equal. Females take longer than males because they defer molting/pupation at each stage in order to grow larger. There is a positive feedback loop between females deferring the molts between instars in order to grow larger, and being able to feed faster in the next instar, thereby growing even faster than males.

At the end of each instar both males and females face alternatives that appear to be determined by the availability of food. If there is too much food, the larva dies. If there is a lot of food, the larva extends growth to continue to grow (probably to some physical or physiological maximum based on maximum size for the exoskeleton, or maximum feeding rate counterbalanced by the cost of maintaining the existing mass, or to some internal physiological state). If there is just the right amount of food, the larva molts or pupates. If there is less than the optimal amount of food, the larva defers molting or pupation to grow larger. If there is not enough food to molt to the next instar or to pupate, the larva enters diapause and waits for more food (and may die eventually). If there is too little food to enter diapause, the larva dies. Females have an additional option; they can change their feeding behavior from actively filtering to retaining particles. This changes the available food for other larvae in the same microcosm. It also appears to be less efficient at extracting nutrients from the food supply and results in slower growth and a smaller size distribution among female larvae.

Males appear to have a minimum size at pupation that is determined by food early in the larval life (around day 3); once they reach this size, they may pupate or continue to feed and grow if food is available. Prime males appear to have a maximum size that is determined by the amount of food before day 6. The non-Prime males are able to grow larger than the Prime male in some circumstances when food is added to the test tubes on day 6 (after the Prime

male has pupated). Females in these experiments do not reach a distinct maximum size; females require more food to pupate at all than males do, and grow larger than males and for longer at all food levels. Females appear to grow to a size commensurate with the food level and require at least 2 mg food/larva in order to pupate. Females appear to have a maximum larval period after starvation that is not affected by the amount of food added to end the period of starvation (in the third experiment). Males reach their maximum size and pupate, while females grow until they reach their maximum larval period and then pupate.

This means that the triggering of pupation is dependent on at least: age or time, mass, environmental food conditions, and internal physiological status. Furthermore, the maximum size of pupae is probably also constrained by early life history due to the size of the head capsule, the feeding apparatus or some other physical or physiological feature determined by the molt to the 4th instar.

## Discussion

Competition is the result of multiple individual actions and interactions, but results in a specific distribution of larval sizes for each set of environmental conditions (in the first experiment: food level, density, aliquot and timespan). For females, at high relative food abundance, the Prime and the Average masses are large and there is a small difference between them. At low relative food abundance, the Prime and Average masses are small and there is a larger difference between them, but not the largest difference. At intermediate relative food abundance, high density favors the Prime female over the others; the Prime female mass is almost as large as at higher food abundances and the Average is similar to that at the low density. The controlling factors appear to be different aspects of the food resource (e.g. total food per test tube after day 4 and food/larva after day 4) as affected by the four factors above. The Prime female grows to a size determined by the total food (after day 4) in the test tubes where the food/larva (after day 4) is 4 mg or higher. Below that food/larva amount, the Prime female grows to a size determined by the food/larva. The age at pupation for the Prime female is determined by the total food at all levels of food/larva. The size at pupation of the Average females is determined by the food/larva at all levels of food/larva.

The relationships in the previous paragraph, taken together, mean that the Prime female outcompetes the non-Prime females for particles when they are most abundant and grows largest, fastest and pupates earliest. Because particles are abundant, the non-Prime females grow large also, but only as large as the food/larva indicates. At intermediate food levels (4 mg food/larva in the first experiment), the higher density results in larger Prime females because the total food is greater, but the Average size of females is the same as in the low density, intermediate competition treatment (also 4 mg food/larva). The high density, 4 mg food/larva results in the largest difference between the Prime and Average females. At lower levels of food/larva, the females switch from active filtering to retention and the sizes of the Prime and Average females are small, but the size distribution (the difference between the Prime and Average masses) is smaller than the high density, 4 mg food/larva test tubes.

The experiment crossed food level and density, which produce 4 different competitive environments, with aliquot and timespan, which alter the timing and amount of food across the experiment. Many Prime females pupate before the final aliquot is added for the 6 day timespan, so the amount of food after the last additions before that, on day 3 for the 3 day timespan treatments, and on day 4 for the 4 aliquot, 6 day timespan treatments, reflect the actual amount of food in the test tubes. Aliquot and timespan interact with FxD separately, with the exception of the 4-way interaction for the non-Prime females. When a large amount of food is added on day 3, the non-Prime females experience a release from competition and grow larger than

expected; this causes a 4-way interaction in the ANOVA for the Average females. Both the 3-way interactions, FxDxT and FxDxA, are significant for Prime and Average female masses. FxD represents the residual effect of food level and density on these variables after the higher order interactions have been removed. The Prime female age and Average female mass are significant in this interaction. Competition extends the time to pupation and reduces the size of the non-Prime females after the effects of aliquot and timespan have been removed. Three other interactions involve density and one or more of the aliquot and timespan treatments; these are aspects of the abundance of food, and both aliquot and timespan interact with competition, so they could also indicate competition from aspects of the food availability independent of the factor, food level. If this were the case, the same pattern of distribution of sizes should appear. There is no evidence that these three interactions, DxAxT, DxA, or DxT, affect competition among females. They appear to describe the growth of females.

Competition among males also results in a pattern of size distributions. Prime males respond to total food and food/larva similarly to Prime females, but the Average males do not correspond to the Average females. Males appear to be competing for a different subset of the yeast particles than females; this is probably related to the different sizes of male and female larvae and their associated feeding structures. Males compete more intensely with other males and females with other females. Females do affect competition among males by reducing the low particle numbers even further due to a change in feeding behavior (retention). Non-Prime males experience a release from competition after the pupation of the Prime male, and also benefit from the day 6 addition of food. The release from competition results in a 0.01 mg increase in size (FxD), while the combined release and extra growth on the day 6 food results in a 0.14 mg increase (FxDxT). Unlike females, males appear to have two life history strategies: early pupation versus taking longer and growing larger. Early pupation may enhance mating success for males, while larger adults live longer and may have more opportunities to mate.

Unlike females, the DxAxT and DxT interactions appear to produce size distributions of males that resemble the FxDxT and FxD competitive outcomes. All of the significant interactions that include density as one of the factors result in outcomes that look like competition among males, even those that do not include total food. The factors that describe the distribution of food in time (aliquot, timespan) interact with density independently of the food level to alter the competitive environment for males, but not for females. Note that the 4-way interaction (FxDxAxT) is not significant for males in the MANOVA or the ANOVAs. This suggests that the mechanism of competition differs between males and females. Unlike females, the aliquot treatment does not interact with FxD to affect competition. This suggests that females switch from actively filtering to retention and back in response to the aliquot treatment, but males are unaffected because they are always actively filtering. There is some evidence that males respond quickly to small changes in the abundance of food, while females respond more slowly or not at all.

Competition results in one or a few winners and multiple losers in each container. The Prime individuals (one male and one female in each microcosm) represent the winners and the Average individuals represent the losers. Intra specific competition among *A. aegypti* larvae is reviewed in [1]. Most of these studies do not take into account the concept of Prime individuals. Similarly, inter specific competition between *A. aegypti* larvae and other mosquito larvae is alleged to determine the abundance and distribution of the species in nature [3, 6, 11–13, 40–62]. None of these studies accounted for differential effects of competition on winners and losers in the microcosms. In the first experiment in this study, the Prime female and the Prime male are affected by the food, density and timespan treatments, but grow and pupate as though they had access to the total food in the microcosm. The non-Prime individuals, as represented by the Average male mass and Average female mass, experience negative effects of competition

and these increase as the competition increases in intensity. Extrapolating to inter specific competitive contests, it is possible that the Prime individuals (of both species) are less affected or not affected by the competitive interactions that the average size and average age at pupation measure. Far from organizing communities as suggested in the introduction, larval competition may be largely irrelevant to the ecology and evolution of mosquitoes; this needs to be examined in further experiments.

The growth of larvae is described by interactions that don't involve density for males, and for females, by interactions that don't involve both food and density. Larvae are expected to grow in a sigmoid growth curve for each instar. The data suggest that males molt and pupate earlier on the growth curves than females, but both sexes alter the timing of the molts based on the availability of food. The first experiment suggests that early food inputs are better than late food inputs, and that timing the input during the 3$^{rd}$ and 4$^{th}$ instars results in larger pupae. The third experiment further extends the observation that early food inputs are better than late food inputs, but also indicates that the larvae replenish internal resources after a period of starvation. Not all the food in the late food inputs appears as mass at pupation, and less of the food ends up in mass at the day 8 delay than the day 6 delay (two days longer starvation).

The first experiment explains such a high percentage of the variance in mass and age at pupation for males and females that it is essentially a deterministic description of the responses of mosquito larvae to their food environment. Natural containers are larger and food inputs are less regular; both these alterations could reduce the deterministic aspect of the larval environment. However, it suggests that the choice of the larval environment is important to the success of each adult female's progeny. To the extent possible ovipositing females should choose optimal containers. Furthermore, females should ideally provision their eggs to result in the largest possible first instar hatchlings. Opposing these forces are the possibility that any one container could be destroyed or drained or just barren of food, and that fewer, larger eggs mean a greater risk of complete reproductive failure in an unpredictable environment. Experiments on container choice, oviposition preference, and propagule size (egg size) and provisioning (egg quality) by females should indicate how deterministic the larval outcomes are in nature.

Females appear to prefer containers that host congeneric larvae, that are larger, and that are darker or shaded [28, 63–67] and that host other species' larvae and pupae [44]. They may be attracted to chemical cues from bacteria in the containers [68–70]. This suggests that viable larvae, a larger container volume, and a protected container are the most important parameters that the ovipositing females search for. Propagule size (egg size) is directly related to female body size and blood meal size, which are also correlated with each other [34, 71, 72]. Some small females produce fewer, larger eggs than expected, suggesting that large eggs have an advantage over small eggs despite the greater risk of reproductive failure with fewer larger eggs. Provisioning (egg quality) does not appear to have been measured.

Mosquito larvae grow differently on different food sources and substrates [3, 6, 7, 42, 43, 45, 73–83]. In addition to the supplied food sources, usually yeast cells or bacteria, bacteria in the gut of the larvae may provide nutrients [84–87] and endoparasites may compete for nutrients [88]. The difference in feeding behavior between male and female larvae suggests that female mosquito larvae may harvest nutrients from retained yeast cells and/or bacteria while male larvae do not.

Ecologically, and evolutionarily, *A. aegypti* mosquitoes may have adapted to survive in impermanent aquatic habitats by growing quickly in response to food inputs, then resisting starvation until the next input, eventually reaching a size that permits pupation as a viable adult. Males and females appear to have different larval life history strategies probably due to their different reproductive roles. Adult males need to find virgin females [35, 89–93] to

inseminate, while adult females need to be as large as possible to produce more eggs and larger eggs [4, 5, 34, 40, 41, 71, 72, 94–103], but see [96]. Size also increases longevity, which probably increases the likelihood of reproductive success. Males pupate earlier than females and at a smaller size in order to inseminate females of their own cohort, but some males aren't able to pupate quickly and delay pupation to grow larger on later additions of food. This bi modal distribution of mass and age at pupation may result in reproductive success for the slower-growing males in subsequent cohorts.

It is also clear that there are physiological conditions that affect the age and mass at pupation [4, 10, 34, 94–95, 104–109]. These pupal triggers are also affected by the food environment in these experiments, but the relationship between the internal state and the external environment is not as clear as the relationship between competition and growth and the pupal mass and age.

## Supporting information

**S1 Dataset. Experiment 1.** Excel file of the mass, age at pupation and sex of each individual pupa by treatment.
(XLSX)

**S2 Dataset. Experiment 2.** Excel file of the mass, age at pupation and sex of each individual pupa by treatment.
(XLSX)

**S3 Dataset. Experiment 3.** Excel file of the mass, age at pupation and sex of each individual pupa by treatment.
(XLSX)

**S1 Fig. Experiment 1.** Scatterplot of mass at pupation (mg) in 0.2 mg increments versus the number of females at each mass.
(DOCX)

**S2 Fig. Experiment 1.** Scatterplot of age at pupation (days) versus the number of females pupating on each day.
(DOCX)

**S3 Fig. Experiment 1.** Scatterplot of mass at pupation (mg) in 0.2 mg increments versus the number of males at each mass.
(DOCX)

**S4 Fig. Experiment 1.** Scatterplot of age at pupation (days) versus the number of males pupating on each day.
(DOCX)

**S5 Fig. Experiment 1.** 3D visualization of Survival for FxDxT.
(DOCX)

**S6 Fig. Experiment 1.** 3D visualization of Prime female mass for FxDxT.
(DOCX)

**S7 Fig. Experiment 1.** 3D visualization of Prime female age for FxDxT.
(DOCX)

**S8 Fig. Experiment 1.** 3D visualization of estimated growth rate for Prime females for FxDxT.
(DOCX)

**S9 Fig. Experiment 1.** 3D visualization of Average female mass for FxDxT.
(DOCX)

**S10 Fig. Experiment 1.** 3D visualization of Prime female mass MINUS Average female mass for FxDxT.
(DOCX)

**S11 Fig. Experiment 1.** 3D visualization of Prime male mass for FxDxT.
(DOCX)

**S12 Fig. Experiment 1.** 3D visualization of Prime male age for FxDxT.
(DOCX)

**S13 Fig. Experiment 1.** 3D visualization of Average male mass for FxDxT.
(DOCX)

**S14 Fig. Experiment 1.** 3D visualization of Prime male mass MINUS Average male mass for FxDxT.
(DOCX)

**S15 Fig. Experiment 1.** 3D visualization of estimated growth rates for both Prime females and Prime males for FxDxT.
(DOCX)

**S16 Fig. Experiment 1.** 3D visualization of Prime female mass MINUS Prime male mass for FxDxT.
(DOCX)

**S17 Fig. Experiment 1.** 3D visualization of Prime female mass for FxDxA.
(DOCX)

**S18 Fig. Experiment 1.** 3D visualization of Average female mass for FxDxA.
(DOCX)

**S19 Fig. Experiment 1.** 3D visualization of Prime female mass MINUS Average female mass for FxDxA.
(DOCX)

**S20 Fig. Experiment 1.** 3D visualization of estimated Prime female growth rate for FxDxA.
(DOCX)

**S21 Fig. Experiment 1.** 3D visualization of Prime female age and Average female mass for FxD.
(DOCX)

**S22 Fig. Experiment 1.** 3D visualization of Prime male age and Average male mass for FxD.
(DOCX)

**S23 Fig. Experiment 1.** 3D visualization of Prime female mass for FxAxT.
(DOCX)

**S24 Fig. Experiment 1.** 3D visualization of Average female mass for FxAxT.
(DOCX)

**S25 Fig. Experiment 1.** 3D visualization of Prime male mass and age for FxA.
(DOCX)

**S26 Fig. Experiment 1.** 3D visualization of Prime female mass and age for FxT.
(DOCX)

**S27 Fig. Experiment 1.** 3D visualization of Prime male mass and age for FxT.
(DOCX)

**S28 Fig. Experiment 1.** 3D visualization of mass versus total food after day 4 for FxT.
(DOCX)

**S29 Fig. Experiment 1.** 3D visualization of Prime female mass and age for AxT.
(DOCX)

**S30 Fig. Experiment 1.** 3D visualization of Prime male mass and age for AxT.
(DOCX)

**S31 Fig. Experiment 1.** 3D visualization of Prime female mass for DxAxT.
(DOCX)

**S32 Fig. Experiment 1.** 3D visualization of Average male mass for DxAxT.
(DOCX)

**S33 Fig. Experiment 1.** 3D visualization of Prime female age at pupation for DxA.
(DOCX)

**S34 Fig. Experiment 1.** 3D visualization of Prime female mass and age for DxT.
(DOCX)

**S35 Fig. Experiment 1.** 3D visualization of Prime male mass and age for DxT.
(DOCX)

**S36 Fig. Experiment 1.** 3D visualization of mass versus average food/larva after day 4 for DxT.
(DOCX)

**S37 Fig. Experiment 3.** Heuristic 3D graph of male and female masses and ages at pupation for the 3-way interaction between second food input, delay and sex in the third experiment.
(DOCX)

**S38 Fig. Experiment 3.** Graph of the wet weight (mg) of male and female pupae against the dry weight of the yeast food source.
(DOCX)

**S1 Table. Experiment 1.** Food input schedule for each treatment according to the factors: food level (16 mg, 32 mg), aliquot (2 portions, 4 portions), and timespan (over days 0 to 3 or days 0 to 6). Density affects the food/larva which is also affected by the food input schedule.
(DOCX)

**S2 Table. Experiment 2.** Treatment numbers with density (larvae/vial), food/larva (2 mg, 3 mg, 4 mg, and 5 mg dry weight of yeast), and number of replicates.
(DOCX)

**S3 Table. Experiment 3.** Treatments showing the amount of the second food input (1 mg, 2 mg, 3 mg dry weight of yeast) and the delay (day of second food input, day 6 or day 8) with the number of replicates. All treatments received 1 mg dry weight of yeast on day 0 of the experiment.
(DOCX)

**S4 Table. Experiment 1.** Significant correlations between composite scores and the 7 dependent variables with MANOVA significance levels and R squared by significant contrasts. (DOCX)

**S5 Table. Experiment 1.** Explanation of the meaning of the single df contrasts. (DOCX)

**S6 Table. Experiment 1.** MANOVA contrasts for competition interactions (food, density, aliquot, timespan: FxDxT, FxDxA, FxD). R squared values, significance levels and discriminant function coefficients by dependent variable for the three interactions. (DOCX)

**S7 Table. Experiment 1.** MANOVA contrasts for interactions involving only attributes of the food supply (food, aliquot, timespan: FxA, FxT, AxT). R squared values, significance levels and discriminant function coefficients by dependent variable for the three interactions. (DOCX)

**S8 Table. Experiment 1.** MANOVA contrasts for interactions between density and the non-food level attributes of the food supply (density, aliquot, timespan: DxAxT, DxT). R squared values, significance levels and discriminant function coefficients by dependent variable for the three interactions. (DOCX)

**S9 Table. Experiment 1.** ANOVA significant r squared values for the main treatments and significant interaction contrasts from the MANOVA for the 7 dependent variables. The row labelled "Subtotal r squared" includes only the r squared values for the contrasts listed in this table; the row labelled "Total r squared" includes all of the r squared values in the respective ANOVAs. (DOCX)

**S10 Table. Experiment 1.** ANOVA mean squares, (r squared), and significance [*] for each single df contrast across the 7 dependent variables. (DOCX)

**S11 Table. Experiment 1.** Means (SE) for the main effects: food, density, aliquot and timespan; for the 7 dependent variables. (DOCX)

**S12 Table. Experiment 1.** Number of replicates (N). (DOCX)

**S13 Table. Experiment 1.** Means (SD) of arcsin transformed percent Survival. (DOCX)

**S14 Table. Experiment 1.** Means (SD) Prime female mass at pupation (mg). (DOCX)

**S15 Table. Experiment 1.** Means (SD) Prime female age at pupation (days). (DOCX)

**S16 Table. Experiment 1.** Means (SD) Average female mass at pupation (mg). (DOCX)

**S17 Table. Experiment 1.** Means (SD) Prime male mass at pupation (mg). (DOCX)

**S18 Table. Experiment 1.** Means (SD) Prime male age at pupation (days).
(DOCX)

**S19 Table. Experiment 1.** Means (SD) Average male mass at pupation (mg).
(DOCX)

**S20 Table. Experiment 1.** Means (SE) for FxDxT for arcsin transformed percent Survival.
(DOCX)

**S21 Table. Experiment 1.** Means (SE) for FxDxT for Prime female mass and age, and Average female mass. Estimated growth rate and difference between the Prime and Average female mass.
(DOCX)

**S22 Table. Experiment 1.** Means (SE) for Prime female mass and age at pupation and Average female mass at pupation for the interaction FxDxT. Total food after day 4 and food/larva after day 4.
(DOCX)

**S23 Table. Experiment 1.** Means (SE) for FxDxT for Prime male mass and age, and Average male mass.
(DOCX)

**S24 Table. Experiment 1.** Means (SD) Average female mass for FxDxAxT.
(DOCX)

**S25 Table. Experiment 1.** Means (SE) for FxDxA for Prime female mass and age, and Average female mass. Estimated growth rate and difference between the Prime and Average female mass.
(DOCX)

**S26 Table. Experiment 1.** Means (SE) for FxDxA for Prime female mass and age, and Average female mass. Total food and food/larva after day 4.
(DOCX)

**S27 Table. Experiment 1.** Means (SE) for FxD for Prime female mass and age, and Average female mass. Expected mean values for Prime female age and Average female mass.
(DOCX)

**S28 Table. Experiment 1.** Means (SE) for FxD for Prime male mass and age, and Average male mass. Expected mean values for Prime male age and Average male mass.
(DOCX)

**S29 Table. Experiment 1.** Means (SE) for FxAxT for Prime female mass and age, and Average female mass. Estimated growth rate and difference between the Prime and Average female mass.
(DOCX)

**S30 Table. Experiment 1.** Means (SE) for FxAxT for Prime female mass and age, and Average female mass. Total food and food/larva after day 4.
(DOCX)

**S31 Table. Experiment 1.** Means (SE) for FxA for Prime male mass and age, and Average male mass. Expected mean values for the Prime male mass and age and the Average male mass.
(DOCX)

**S32 Table. Experiment 1.** Means (SE) for Prime female mass and Average female mass at pupation for the interaction FxA. Differences between the Prime female mass and the Prime male mass and between the Average female mass and the Average male mass.
(DOCX)

**S33 Table. Experiment 1.** Means (SE) for arcsin transformed percent Survival for the interaction FxT.
(DOCX)

**S34 Table. Experiment 1.** Means (SE) for Prime female mass and age and Average female mass for the interaction FxT. Total food, expected values, growth rates and the differences between Prime and Average female masses.
(DOCX)

**S35 Table. Experiment 1.** Means (SE) for Prime male mass and age and Average male mass for the interaction FxT. Expected values, growth rates and the differences between Prime and Average female masses.
(DOCX)

**S36 Table. Experiment 1.** Means (SE) for Prime male mass and age and Average male mass for the interaction FxT. Differences between the Prime female mass and Prime male mass and between the Average female mass and the Average male mass.
(DOCX)

**S37 Table. Experiment 1.** Means (SE) for Prime female mass and age at pupation and Average female mass at pupation for the interaction AxT. Expected values, growth rates and the differences between the Prime female mass and the Average female mass.
(DOCX)

**S38 Table. Experiment 1.** Means (SE) for Prime male mass and age at pupation and Average male mass at pupation for the interaction AxT. Expected values, growth rates and the differences between the Prime male mass and the Average male mass.
(DOCX)

**S39 Table. Experiment 1.** Means (SE) for Prime female mass for the interaction AxT. Differences between Prime female mass and Average female mass, Prime male mass and Average male mass, Prime female mass and Prime male mass, and Average female mass and Average male mass. Total food after day 4 and food/larva after day 4.
(DOCX)

**S40 Table. Experiment 1.** Means (SE) for Prime female mass and age at pupation and Average female mass at pupation for the interaction DxAxT. Estimated growth rates and the differences between the Prime female mass and the Average female mass. Total food after day 4 and food/larva after day 4.
(DOCX)

**S41 Table. Experiment 1.** Means (SE) for Prime male mass and age at pupation and Average male mass at pupation for the interaction DxAxT. Estimated growth rates and the differences between the Prime male mass and the Average male mass.
(DOCX)

**S42 Table. Experiment 1.** Means (SE) for Prime female mass and Average male mass for the interaction DxAxT. Prime female mass MINUS Average female mass, Prime female mass

MINUS Prime male mass, Prime male mass MINUS Average male mass, Average female mass MINUS Average male mass.
(DOCX)

**S43 Table. Experiment 1.** Means (SE) for Prime female mass and age at pupation and Average female mass at pupation for the interaction DxA. Estimated growth rates, differences between Prime and Average female masses, expected mean values for Prime female age, and food levels after day 4 and on day 0.
(DOCX)

**S44 Table. Experiment 1.** Means (SE) for arcsin transformed percent Survival for the interaction DxT. Expected values.
(DOCX)

**S45 Table. Experiment 1.** Means (SE) for Prime female mass and age at pupation and Average female mass at pupation for the interaction DxT. Expected mean values, growth rates, differences between Prime and Average female masses, food levels.
(DOCX)

**S46 Table. Experiment 1.** Means (SE) for Prime male mass and age at pupation and Average male mass at pupation for the interaction DxT. Expected mean values, growth rates, differences between Prime and Average male masses.
(DOCX)

**S47 Table. Experiment 1.** Means (SE) for Prime female mass at pupation for the interaction DxT. Differences between Prime and Average female masses, Prime and Average male masses, Prime female and Prime male masses, and Average female and male masses. Food/larva.
(DOCX)

**S48 Table. Experiment 2.** Survival of larvae across low densities showing sex ratio distortion at the lowest density.
(DOCX)

**S49 Table. Experiment 2.** Mass at pupation and age at pupation for males and females at low densities.
(DOCX)

**S50 Table. Experiment 3.** Means (SD) for mass and age at pupation by treatment factors.
(DOCX)

**S51 Table. Experiment 3.** Counts and totals for larval deaths by treatment and time period.
(DOCX)

**S52 Table. Experiment 3.** Mean (SD) age at death (days) of larvae by treatment.
(DOCX)

**S53 Table. Experiment 3.** Means (SE) for mass and age at pupation for main effects: food input amount, day of second food input, and sex.
(DOCX)

**S54 Table. Experiment 3.** MANOVA discriminant function coefficients, R squared values and P values for the single df contrasts.
(DOCX)

**S55 Table. Experiment 3.** Mean squares (MS), r squared values and P values for the ANOVAs for the two variables mass and age at pupation for each single df contrast.
(DOCX)

**S56 Table. Experiment 3.** Means (SE) for mass (mg) for the interaction food 2 x delay x sex.
(DOCX)

**S57 Table. Experiment 3.** Means (SE) for age (days) for the interaction food 2 x delay x sex.
(DOCX)

**S58 Table. Experiment 3.** Means (SE) for estimated growth rates (mg/day) for the interaction food 2 x delay x sex.
(DOCX)

**S59 Table. Experiment 3.** Means (SE) for mass (mg) for the interaction food 1 x delay x sex.
(DOCX)

**S60 Table. Experiment 3.** Means (SE) for age (days) for the interaction food 1 x delay x sex.
(DOCX)

**S61 Table. Experiment 3.** Means (SE) for estimated growth rates (mg/day) for the interaction food 1 x delay x sex.
(DOCX)

**S62 Table. Experiment 3.** Means (SE), expected values and differences for mass (mg) for the interaction food 1 x delay.
(DOCX)

**S63 Table. Experiment 3.** Means (SE), expected values and differences for age (days) for the interaction food 1 x delay.
(DOCX)

**S64 Table. Experiment 3.** Means (SE), expected values and differences for mass (mg) for the interaction food 2 x delay.
(DOCX)

**S65 Table. Experiment 3.** Means (SE) for age (days) for the interaction food 2 x delay (not significant in the ANOVA).
(DOCX)

**S66 Table. Experiment 3.** Means (SE), expected values and differences for age (days) for the interaction food 1 x sex.
(DOCX)

**S67 Table. Experiment 3.** Means (SE) for mass (mg) for the interaction food 1 x sex (not significant in the ANOVA).
(DOCX)

**S68 Table. Experiment 3.** Means (SE) for estimated growth rates (mg/day) for the interaction food 1 x sex.
(DOCX)

**S69 Table. Experiment 3.** Means (SE), expected values and differences for age (days) for the interaction food 2 x sex.
(DOCX)

**S70 Table. Experiment 3.** Means (SE) for mass (mg) for the interaction food 2 x sex (not significant in the ANOVA).
(DOCX)

**S71 Table. Experiment 3.** Means (SE) for estimated growth rates (mg/day) for the interaction food 2 x sex.
(DOCX)

**S1 Text. Detailed methods.**
(DOCX)

**S2 Text. Summary of results.**
(DOCX)

**S3 Text. Experiment 1.** First experiment results analysis by interaction.
(DOCX)

**S4 Text. Experiment 1.** First experiment results summary by interaction.
(DOCX)

**S5 Text. Experiment 3.** Third experiment analysis of individual ANOVA contrasts.
(DOCX)

## Acknowledgments

I dedicate this paper to the memory of Dr. Joanne M. Werts; she did not live to see the completed document. I would like to thank Dr. J. H. Frank for his advice and encouragement. Drs. W. Bradshaw, J. R. Linley, L. P. Lounibos, G. F. O'Meara, and J. M. Werts also read the manuscript and offered helpful suggestions. I am grateful to Dr. Werts and the Department of Zoology at Duke University for computer time. My late partner, Richard O. Merritt, provided posthumous support to allow me to complete the paper after many years delay. My husband, Dr. Douglas Kline, provided encouragement, editing and technical guidance in the completion of the task.

## Author Contributions

**Conceptualization:** Kurt Steinwascher.

**Data curation:** Kurt Steinwascher.

**Formal analysis:** Kurt Steinwascher.

**Funding acquisition:** Kurt Steinwascher.

**Investigation:** Kurt Steinwascher.

**Methodology:** Kurt Steinwascher.

**Project administration:** Kurt Steinwascher.

**Resources:** Kurt Steinwascher.

**Software:** Kurt Steinwascher.

**Supervision:** Kurt Steinwascher.

**Validation:** Kurt Steinwascher.

**Visualization:** Kurt Steinwascher.

**Writing – original draft:** Kurt Steinwascher.

**Writing – review & editing:** Kurt Steinwascher.

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
