## [Decision Letter · Decision Letter 0]

29 Jul 2020

PONE-D-20-16489

Competition and growth among *Aedes aegypti* larvae: effects of distributing food inputs over time

PLOS ONE

Dear Dr. Steinwascher,

Thank you for submitting your manuscript to PLOS ONE. After careful consideration, we feel that it has merit but does not fully meet PLOS ONE’s publication criteria as it currently stands. Therefore, we invite you to submit a revised version of the manuscript that addresses the points raised during the review process.

We look forward to receiving your revised manuscript.

Kind regards,

Jiang-Shiou Hwang, Ph.D.

Academic Editor

PLOS ONE

Journal Requirements:

2. Please amend either the title on the online submission form (via Edit Submission) or the title in the manuscript so that they are identical.

Reviewers' comments:

Reviewer's Responses to Questions

**Comments to the Author**

1. Is the manuscript technically sound, and do the data support the conclusions?

Reviewer #1: Partly

Reviewer #2: Yes

2. Has the statistical analysis been performed appropriately and rigorously? 

Reviewer #1: I Don't Know

Reviewer #2: Yes

3. Have the authors made all data underlying the findings in their manuscript fully available?

Reviewer #1: Yes

Reviewer #2: Yes

4. Is the manuscript presented in an intelligible fashion and written in standard English?

Reviewer #1: No

Reviewer #2: Yes

5. Review Comments to the Author

Reviewer #1: There are too many raw data and too detail unnecessary duplicated information and the manuscript needs to have more concentrations on the major objectives. The paper is too long with many unnecessary descriptions. It is difficult for readers to catch the main ideas and what is the main findings after carefully reading for a couple of times. The paper needs to be rewritten and follow the PLoS one format.

Reviewer #2: How authors determined the food competition among different life stages ( 1st to 4th instar) of Aedes population during experiment conditions. Have they synchronized the colony? Explain

What are behavioural changed were observed while shortage of food among both male and female population ?

6. PLOS authors have the option to publish the peer review history of their article (what does this mean?). If published, this will include your full peer review and any attached files.

Reviewer #1: No

Reviewer #2: No

---

## [Author Response · Author response to Decision Letter 0]

29 Aug 2020

Reviewer’s comments: (none)

Reviewer’s Responses to Questions

Comments to the Author

1. Is the manuscript technically sound, and do the data support the conclusions?

Reviewer #1: Partly

Reviewer #2: Yes

Reviewer #1 answered “Partly” to this question. I disagree. I believe that the experiments described in the manuscript were conducted rigorously, with appropriate controls, replication, and sample sizes. I had input from both mosquito researchers and statisticians to design the experiments and I was supervised during the running of the experiments by my post-doctoral advisor and observed by other colleagues. I assert that the high level of explained variance (greater than 80% for the mass variables and greater than 60% for the age variables in the first and third experiments) is an indication that the experiments themselves were well designed and well executed. I discuss the statistical results meticulously in the supporting information (now S3 Text for the first experiment, and S5 Text for the third experiment) and follow standard analytical techniques to determine the numerical explanation for the significance of the results. I then interpret the statistically significant results according to the biology of mosquitoes as understood by me and other researchers (please see my earlier paper for a full review of the results of other researchers). Finally, in S2 Text, I condense these results into the 16 novel findings that are listed in the abstract; these 16 novel findings now organize the results section in the main manuscript.

2. Has the statistical analysis been performed appropriately and rigorously?

Reviewer #1: I Don’t Know

Reviewer #2: Yes

Reviewer #1 answered “I don’t know” to this question. I appreciate the candor. The statistical treatments were determined before the experiments were conducted and are integral to the experimental designs. In the case of the second experiment, where the outcome deviated from the assumptions of normality, I did not perform the original statistical analysis, or any statistical analysis, but reported the deviation from normality as an observation. The failure of a large, replicated experiment can be instructive, particularly in designing other experiments to investigate the circumstances surrounding the failure. In this body of work, I used the failure of the second experiment to design the third experiment. In the first and third experiments, the outcomes were analyzed appropriately based on the experimental designs described in the methods section and reported in detail in the supporting information.

3. Have the authors made all data underlying the findings in their manuscript fully available?

Reviewer #1: Yes

Reviewer #2: Yes

The three Excel files of the raw data are included as supporting information.

4. Is the manuscript presented in an intelligible fashion and written in standard English?

Reviewer #1: No

Reviewer #2: Yes

Reviewer #1 answered “No” to this question. There were no specific issues listed, but there were comments to the author in section 5.

5. Review Comments to the Author

Reviewer #1: There are too many raw data and too detail unnecessary duplicated information and the manuscript needs to have more concentration on the major objectives. The paper is too long with any unnecessary description. It is difficult for readers to catch the main ideas and what is the main findings after carefully reading for a couple of times. The paper needs to be rewritten and follow the PLoS one format.

I take full responsibility for this criticism. All of my pre-submission reviewers were familiar with the experimental design and with mosquito biology, with two exceptions. The two pre-submission reviewers that did not have prior knowledge of the experiment caused me to add more detail about the mosquito larval life cycle and to explain the experimental analysis and statistical treatment more clearly. The comments by reviewer #1 showed me that I needed to explain my points better, and I rewrote the entire Results section to focus on the 16 major novel findings about Aedes aegypti larval biology. Although this reorganization of the content results in some further duplication because the statistical results support multiples of the 16 findings and have to be referenced under each of them, I think that the manuscript is much improved by the reorganization. 

I am open to further modifications if there are outstanding issues that I haven’t addressed. I strongly believe that peer review improves the quality of manuscripts and of science in general.

Reviewer #2: How authors determined the food competition among different life stages (1st-4th instar) of Aedes population during experiment conditions. Have they synchronized the colony? Explain. What are behavioural changed were observed while shortage of food among both male and female population?

These are great questions.

How authors determined the food competition among different life stages (1st-4th instar) of Aedes population during experiment conditions. 

Food competition among mosquito larvae has been the subject of many papers. I published an earlier paper in PLoS ONE (Steinwascher, 11 15 2018) that describes food competition in great detail. In the current manuscript, I used the results of the prior paper to set up a 2x2 grid that resulted in four competitive environments (food x density) and crossed this with 2 other factors, aliquot and timespan, that affected the availability of food on a daily basis. It isn’t possible to measure the trajectory of growth of the larvae without handling them, and it isn’t possible to sex the larvae visually until the very end of the 4th instar or pupation. I used pupation to measure the outcome of growth of the larvae and calculated 7 dependent variables for each replicate (survival, pupal masses and ages of males and females). The 7 dependent variables, plus some post analytic comparisons (growth rate, size distributions) allowed me to infer the effects of the independent factors on the food competition and growth processes that occurred in the microcosms. I could not measure the growth of the early instars and the timing of the molts from 1st to 2nd, etc. However, growth is generally accepted to resemble a sigmoid curve when mass is plotted against time and this model fits the data that I observed. The mechanics of filter feeding have been studied by Karen Porter among others, in the 1970s, and although I did not measure the filtering ability directly, the results of their experiments fit the data that I observed.

Have they synchronized the colony? Explain. 

I fed the colony on a mouse or chick approximately once a week over several months. There was always an oviposition site available and the filter papers from the oviposition jars were changed once or twice a week. These filter papers with eggs attached were stored in an insectary until it was time to regenerate the colony or to conduct an experiment. They were then hatched and used for an experiment; the extra larvae were raised in batch trays and released into a new colony cage. This procedure was to make sure that I had enough eggs to hatch for each experiment, but it effectively resulted in synchronizing the colony. I have added a description of this to the expanded Methods section in the supporting information (S1 Text).

What are behavioural changed were observed while shortage of food among both male and female population?

I couldn’t tell a male larva from a female larva in the microcosms. I also couldn’t tell individual larvae apart except for their size during the 4th instar. The behavior of the larvae in the microcosms was not obviously different across treatments. There was usually an amount of yeast on the bottom of the microcosm and larvae would spend time there, presumably feeding. Other larvae would float in the water column or at the surface. I didn’t see anything that looked like physical interaction, but the larvae wriggle and may contact one another in so doing. I had no way to measure these interactions. Other people have categorized the larval activity and done time/motion studies on them. I reviewed some of this work in my prior paper.

---

## [Decision Letter · Decision Letter 1]

11 Sep 2020

PONE-D-20-16489R1

Competition and growth among *Aedes aegypti* larvae: effects of distributing food inputs over time

PLOS ONE

Dear Dr. Steinwascher,

Thank you for submitting your manuscript to PLOS ONE. After careful consideration, we feel that it has merit but does not fully meet PLOS ONE’s publication criteria as it currently stands. Therefore, we invite you to submit a revised version of the manuscript that addresses the points raised during the review process.

We look forward to receiving your revised manuscript.

Kind regards,

Jiang-Shiou Hwang, Ph.D.

Academic Editor

PLOS ONE

Reviewers' comments:

Reviewer's Responses to Questions

**Comments to the Author**

1. If the authors have adequately addressed your comments raised in a previous round of review and you feel that this manuscript is now acceptable for publication, you may indicate that here to bypass the “Comments to the Author” section, enter your conflict of interest statement in the “Confidential to Editor” section, and submit your "Accept" recommendation.

Reviewer #1: All comments have been addressed

2. Is the manuscript technically sound, and do the data support the conclusions?

Reviewer #1: Yes

3. Has the statistical analysis been performed appropriately and rigorously? 

Reviewer #1: Yes

4. Have the authors made all data underlying the findings in their manuscript fully available?

Reviewer #1: Yes

5. Is the manuscript presented in an intelligible fashion and written in standard English?

Reviewer #1: Yes

6. Review Comments to the Author

Reviewer #1: The author did well for the revision. However, there are still too many references for this kind of study report.

7. PLOS authors have the option to publish the peer review history of their article (what does this mean?). If published, this will include your full peer review and any attached files.

Reviewer #1: No

---

## [Author Response · Author response to Decision Letter 1]

14 Sep 2020

September 14, 2020

Re: PONE-D-20-16489R1. Competition and growth among Aedes aegypti larvae: effects of distributing food inputs over time

Dear Dr. Hwang and reviewers,

Thank you for the opportunity to further improve my submission to PLoS ONE.

I have addressed the single comment by reviewer 1 in section 6. Review Comments to the Author below.

I have also reviewed the manuscript itself to improve grammar and clarity further.

Please let me know if there are any further concerns. I look forward to hearing from you.

Sincerely;

Kurt Steinwascher

Reviewers' comments:

Reviewer's Responses to Questions

Comments to the Author

1. If the authors have adequately addressed your comments raised in a previous round of review and you feel that this manuscript is now acceptable for publication, you may indicate that here to bypass the “Comments to the Author” section, enter your conflict of interest statement in the “Confidential to Editor” section, and submit your "Accept" recommendation.

Reviewer #1: All comments have been addressed

2. Is the manuscript technically sound, and do the data support the conclusions?

Reviewer #1: Yes

3. Has the statistical analysis been performed appropriately and rigorously? 

Reviewer #1: Yes

4. Have the authors made all data underlying the findings in their manuscript fully available?

Reviewer #1: Yes

5. Is the manuscript presented in an intelligible fashion and written in standard English?

Reviewer #1: Yes

6. Review Comments to the Author

Reviewer #1: The author did well for the revision. However, there are still too many references for this kind of study report.

Reviewer #1 says that there are too many references. I disagree with this statement.

This is a research article rather than a review of the field, so I understand the concern that reviewer #1 is expressing. The articles cited represent studies that I found pertinent to my own results whether they supported or disagreed with my results and conclusions. I cite them because the authors deserve credit for their research and findings and their own opinions, whether they agree with my conclusions or not. I did not dissect the experimental designs, the analyses, or the logic behind the conclusions in these articles, which is what would be appropriate for a review of the field. 

There are 109 citations in the article. I examined each one and they fall into largely discreet categories:

1. Prior work that leads to the current study—references 1-33 (total 33). The prior work and the supporting articles about the methods (below) are important and necessary to the foundation of the research question and the experimental design. Some of these studies go back to the 1960s and are not available electronically, but they still represent important work in the field and the authors deserve credit for identifying the questions. 

2. Supporting articles about the methods—references 34-38 (total 5). (see above)

3. Studies about inter specific competition among mosquito larvae—references 40-62 (total 23). The experiments themselves are about intra specific competition (discussed in [1]), but also relate to inter specific competition. I think it is noteworthy that none of the articles on inter specific competition among Aedes mosquito larvae measure the evolutionary and ecological winners of this competition, but rather measure the average size and age at pupation, which represents the losers of the competition. This is the second largest group of articles after the prior work. 

4. Studies on oviposition preference among adult females—reference 63-72 (total 10). The results of the three experiments suggest the relationship between the larval environment and the oviposition preferences among adult females. 

5. Studies on different food sources and their effect on mosquito larval growth—references 73-88 (total 16). The studies on food sources and growth contribute to the understanding of the results of the three experiments by providing an expanded context of food sources, specifically the implication that bacteria in the gut of the larvae may influence nutrition. 

6. Studies on aspects of the life history strategy of adults and larvae—references 39, 89-103 (total 16). The studies on life history strategies contribute to the understanding of the results of the three experiments in the context of the environmental, ecological and evolutionary biology of the mosquitoes. 

7. Studies on physiological triggers of pupation in mosquito larvae—references 104-109 (total 6). The physiological triggers of pupation were not originally something that I thought was important, but the results indicate that it is necessary to take these into consideration. There appears to be no other explanation for differences in the size of pupae after different periods of starvation late in the larval period.

In addition, there is a large body of work on intra specific competition among mosquito larvae that I do not cite in this paper because I cited it and commented on it in a prior paper [1].

In reviewing each of the 109 references I do not find any that should not reasonably be included as references in this paper. I am willing to reconsider this if reviewer #1 has specific issues with one or more of the references.

---

## [Decision Letter · Decision Letter 2]

18 Sep 2020

Competition and growth among *Aedes aegypti* larvae: effects of distributing food inputs over time

PONE-D-20-16489R2

Dear Dr. Steinwascher,

We’re pleased to inform you that your manuscript has been judged scientifically suitable for publication and will be formally accepted for publication once it meets all outstanding technical requirements.

Kind regards,

Jiang-Shiou Hwang, Ph.D.

Academic Editor

PLOS ONE

Additional Editor Comments (optional):

Reviewers' comments:

Reviewer's Responses to Questions

**Comments to the Author**

1. If the authors have adequately addressed your comments raised in a previous round of review and you feel that this manuscript is now acceptable for publication, you may indicate that here to bypass the “Comments to the Author” section, enter your conflict of interest statement in the “Confidential to Editor” section, and submit your "Accept" recommendation.

Reviewer #1: All comments have been addressed

2. Is the manuscript technically sound, and do the data support the conclusions?

Reviewer #1: Yes

3. Has the statistical analysis been performed appropriately and rigorously? 

Reviewer #1: Yes

4. Have the authors made all data underlying the findings in their manuscript fully available?

Reviewer #1: Yes

5. Is the manuscript presented in an intelligible fashion and written in standard English?

Reviewer #1: Yes

6. Review Comments to the Author

Reviewer #1: The revised manuscript looks pretty good for me and it is in a good shape. It is ready to be published. Thanks for the revision and the revision is stratify.

7. PLOS authors have the option to publish the peer review history of their article (what does this mean?). If published, this will include your full peer review and any attached files.

Reviewer #1: No

---

## [Editor Report · Acceptance letter]

23 Sep 2020

PONE-D-20-16489R2 

Competition and growth among *Aedes aegypti* larvae: effects of distributing food inputs over time 

Dear Dr. Steinwascher:

I'm pleased to inform you that your manuscript has been deemed suitable for publication in PLOS ONE. Congratulations! Your manuscript is now with our production department. 

Kind regards, 

on behalf of

Prof. Jiang-Shiou Hwang 

Academic Editor

PLOS ONE